# Phosphoribosylpyrophosphate synthetase as a metabolic valve advances *Methylobacterium*/*Methylorubrum* phyllosphere colonization and plant growth

Cong Zhang[1,2,3,15], Di-Fei Zhou[1,2,3,15], Meng-Ying Wang[1,2,3,15], Ya-Zhen Song[1,2,3,15], Chong Zhang[4,5], Ming-Ming Zhang[1,2,3], Jing Sun[1,2,3], Lu Yao[6], Xu-Hua Mo[1,2,3], Zeng-Xin Ma[1,2,3], Xiao-Jie Yuan[1,2,3], Yi Shao[1,2,3], Hao-Ran Wang[1,2,3], Si-Han Dong[1,2,3], Kai Bao[7], Shu-Huan Lu[8], Martin Sadilek[9], Marina G. Kalyuzhnaya [●][10], Xin-Hui Xing[4,5,11,12] & Song Yang [●][1,2,3,13,14] [✉]

The proficiency of phyllosphere microbiomes in efficiently utilizing plant-provided nutrients is pivotal for their successful colonization of plants. The methylotrophic capabilities of *Methylobacterium*/*Methylorubrum* play a crucial role in this process. However, the precise mechanisms facilitating efficient colonization remain elusive. In the present study, we investigate the significance of methanol assimilation in shaping the success of mutualistic relationships between methylotrophs and plants. A set of strains originating from *Methylorubrum extorquens* AM1 are subjected to evolutionary pressures to thrive under low methanol conditions. A mutation in the phosphoribosylpyrophosphate synthetase gene is identified, which converts it into a metabolic valve. This valve redirects limited C1-carbon resources towards the synthesis of biomass by up-regulating a non-essential phosphoketolase pathway. These newly acquired bacterial traits demonstrate superior colonization capabilities, even at low abundance, leading to increased growth of inoculated plants. This function is prevalent in *Methylobacterium*/*Methylorubrum* strains. In summary, our findings offer insights that could guide the selection of *Methylobacterium*/*Methylorubrum* strains for advantageous agricultural applications.

Numerous microbes in nature inhabit carbon nutrient-scarce environments characterized by low rates of reproduction and growth[1]. Within the phyllosphere, the aerial parts of plants dominated by leaves, flowers, and stems, microbes engage in fierce competition for limited resources while concurrently promoting plant growth or providing protection against biotic and abiotic stressors[2]. Despite these observations, the mechanisms governing the colonization of plants by growth-promoting microbes remain inadequately defined, posing a

significant constraint in the development of probiotic supplements for sustainable agriculture[3–6].

*Methylobacterium*/*Methylorubrum* stands out as a well-studied phyllosphere microbe with documented capabilities to enhance plant growth and yield. These capabilities include the synthesis of auxin, gibberellin, cytokinin, and 1-aminocyclopropane-1-carboxylate (ACC) deaminase, as well as nitrogen fixation[7–9]. Most recognized *Methylobacterium*/*Methlorubrum* cultures are characterized as

A full list of affiliations appears at the end of the paper. [✉]e-mail: yangsong1209@163.com; SongYang@qau.edu.cn

facultative alphaproteobacterial methylotrophs, and it has been hypothesized that their methylotrophic abilities play a crucial role in colonization success[10,11]. Methanol is a one-carbon volatile compound that is released from the methoxy groups of plant structural compounds during both plant growth and decay[12]. The estimated annual contribution of methanol emission is approximately 100 Tg annually[13]. However, concentrations of plant-derived methanol exhibit a diurnal cycle varying from trace amounts to tens of millimoles, depending on the plant's growth stages[12,14,15]. Currently, the cellular determinants enabling the bacterial efficient utilization of plant-derived nutrients, specifically methanol, in the phyllosphere remain poorly defined.

Previous research has established that upon plant colonization, various membrane transporters and efflux systems are induced in plant associated microbes to facilitate the acquisition of limited nutrients[3,4,16,17]. Indeed, the uncharged small molecule, methanol, along with the pyrroloquinoline quinone-dependent methanol dehydrogenase enzyme system, which is situated in the periplasmic space of *Methylobacterium/Methylorubrum*[18], suggests the potential that methanol transport might not be obligatory for methanol utilization[19]. In this context, recent efforts have primarily focused on identifying a set of specific methylotrophic traits in the phyllosphere colonizing *Methylobacterium/Methylorubrum*[20–22]. It has been reported that *Methylobacterium/Methylorubrum* species possess chemotaxis proteins towards methanol, facilitating their congregation in phyllosphere niches such as stomata and hydathodes, thereby ensuring survival and colonization[23]. To facilitate the efficient oxidation of methanol, *Methylorubrum extorquens* PA1, a representative of the *Arabidopsis thaliana* leaf microbiota, enhances the expression of rare earth element-dependent methanol dehydrogenases and corresponding chelators for the uptake of metal ion[24]. In response to fluctuations in intracellular methanol levels, *Mr. extorquens* utilizes specific regulatory proteins to detect toxic formaldehyde, which is a direct oxidative product of methanol. This mechanism allows for the modulation of methylotrophic balance and bacterial growth, and ensures a smooth transition from heterotrophy to methylotrophy[25]. Despite these insight findings contributing to our understanding of microbial processes required for establishment in the phyllosphere, the identification of cellular factors and pathways enhancing central methylotrophic efficiency under methanol limitation remains a major obstacle. This challenge is partly due to the involvement of complex metabolic and regulatory networks[26]. The discovery of such methylotrophic traits represents a crucial step towards developing plant growth-promoting microbial supplements with robust colonization abilities and predictable performance. This advancement will ultimately benefit sustainable agriculture and one-carbon sequestration efforts.

In this investigation, we delve the trade-offs between different carbon assimilation pathways as a crucial factor influencing carbon-utilization efficiency (Fig. 1). Through this exploration, we unveil innovative metabolic combinations that propel the colonization of *Methylobacterium/Methylorubrum* within the phyllosphere. These distinctive combinations played a pivotal role in significantly enhancing plant growth and yield, both in growth chamber environments and practical agricultural solar greenhouses. Notably, even when applied via foliar spray at low abundance, these metabolic combinations continued to positively impact plant development. Our study established unprecedented connections between cellular metabolism and cell growth, shedding light on *Methylobacterium/Methylorubrum* traits that have substantial beneficial effects on plant growth and yield. This breakthrough not only enriches our understanding of the intricate interplay within microbial metabolism but also paves the way for the development of innovative strategies to optimize plant-microbe interactions for enhanced agricultural productivity.

## Results

### Selection of *Mr. extorquens* traits with improved growth under methanol limitation

To gain a deeper understanding of the trade-offs between catabolism and anabolism, a strategy to enhance methanol assimilation was devised as follows. Traits from *Mr. extorquens* AM1 that implement the ribulose monophosphate pathway (RuMP), recognized as a more efficient metabolic route for carbon assimilation compared to the native serine cycle, were incorporated[27]. We engineered the methylotrophic chassis in *Mr. extorquens* AM1 by deleting the *hprA* gene, responsible for encoding a hydroxypyruvate reductase (HPR) in the native serine cycle, resulting in the complete abolition of growth on methanol[28]. Simultaneously, genes of the RuMP cycle from *Bacillus methanolicus* MGA3, namely *hps* and *phi*[29,30], were introduced by overexpressing them on plasmid pCM80 (Supplementary Fig. 1a). The resulting strain, named RS03, initially exhibited minimal growth under standard cultivation conditions (i.e., 120 mM methanol, Supplementary Fig. 1b). In this study, standard cultivation conditions referred to methanol at a concentration of approximately 120 mM, which were clearly above growth-limiting concentrations (15 mM or less). A series of adaptive laboratory evolution (ALE) studies were carried out using short-term stationary phase (STSP) evolutionary methodology (Supplementary Fig. 2, described in Methods). This approach led to the selection of eight evolved strains named RSE01 to RSE08 isolated on days 16 (RSE01) and 25 (RSE02 to RSE08) (Supplementary Fig. 3, Supplementary Table 1). These strains exhibited robust growth under standard cultivation conditions due to the activation of a hypothetical protein (META1_3141) that partially fulfilled the function of HPR in all evolved strains (Supplementary Fig. 4). Intriguingly, these strains did not rely on the RuMP cycle for methanol assimilation (Supplementary Fig. 5). The function of META1_3141 was determined through an in vitro enzymatic assay. The growth fitness of the RSE and SE strains without the plasmid carrying the RuMP genes was evaluated at low methanol concentrations ranging from 5 mM to 15 mM, similar to those found in the plant phyllosphere as previously described[14,15]. All RSE and SE strains displayed slow growth when exposed to 5 mM, 10 mM, or 15 mM methanol (Supplementary Fig. 5).

Subsequently, the serine cycle pathway was restored in evolved *Mr. extorquens* AM1 strains to generate SE-*hprA* strains, where *hprA* was reintroduced to the SE strains via overexpression on plasmid pCM80. In comparison to the control strain AM1$^{WT}$01.1 (AM1$^{WT}$-Δ*hprA*-pCM80:*hprA*), the SE-*hprA* strain (SE01.1) consistently exhibited growth advantages when transitioning from either standard or low methanol concentrations to low methanol concentrations in liquid media. This growth advantage was also evident when the strains were spread on agar plates with low methanol concentrations (Fig. 2a, Supplementary Fig. 6a–c). In contrast, other SE-*hprA* strains, such as SE02.1 and SE07.1, displayed a growth trend similar to the AM1$^{WT}$01.1 strain (Supplementary Fig. 6d).

### Partial loss of PRS function increases *Mr. extorquens* growth under methanol limitation

To pinpoint the pathway facilitating the growth of the SE01.1 strain on low methanol concentrations, we conducted re-sequencing of the evolved strains RSE01, RSE02, and RSE07 (Fig. 2b and Supplementary Table 2). A distinctive single-nucleotide polymorphism (SNP) mutation was identified in the coding sequence of putative phosphoribosylpyrophosphate synthetase (PRS)-*prs* (META1_4249, G$_{112}$ to A$_{112}$, resulting in D38N) in the RSE01 strain (from which SE01.1 strain was derived) (Fig. 2b). Introducing the *prs$^{WT}$* into the *prs$^{EVO}$* allele in the SE01.1 strain, resulted in the formation of the strain SE01.3, which no longer displayed any growth advantage compared to the control strain AM1$^{WT}$01.1 (Fig. 2a). It is essential to note that PRS catalyzes the first rate-limiting reaction, converting ribose-5-phosphate (Ri5P) to 5-phosphoribosyl-1-pyrophosphate (PRPP), a critical step in nucleotide

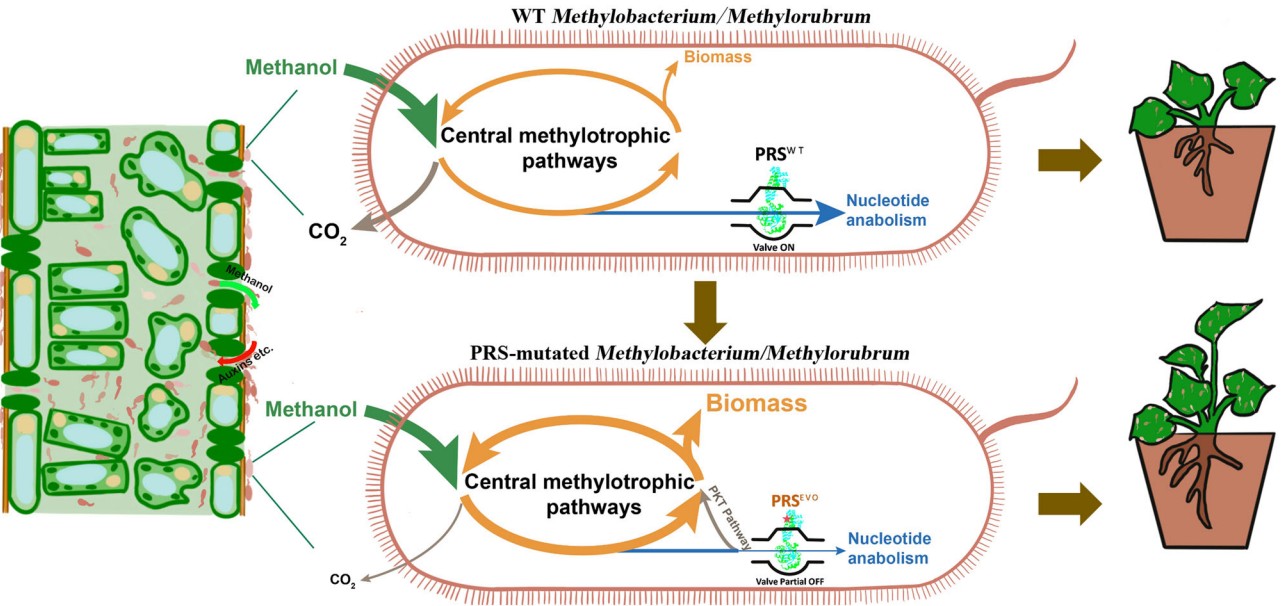

**Fig. 1 | Discovery and analysis of PRS as a metabolic valve capable of enhancing the colonization of *Methylobacterium*/*Methylorubrum* in the phyllosphere and promoting plant growth and yield.** PKT pathway phosphoketolase pathway, PRS[WT] wild-type PRS, PRS[EVO] evolutionary mutant PRS.

biosynthesis pathways[31]. Additionally, ribulose-5-phosphate (Ru5P), a metabolite interconvertible with Ri5P, is catalyzed by another enzyme, phosphoribulokinase, to produce ribulose-1,5-bisphosphate (RuBP). Previous studies have demonstrated the pivotal role of RuBP in the regulation of one-carbon assimilation in *Mr. extorquens* PA1[26].

To comprehend how the mutation from an aspartic acid to an asparagine residue (D38N) affected PRS activity, we overexpressed *prs[WT]* and *prs[EVO]* in *E. coli* BL21 (DE3) and purified the proteins. The specific activities of PRS[WT] and PRS[EVO] were then compared (Fig. 2c and Supplementary Fig. 7). Our analysis revealed that the activity of the heterologously expressed PRS[EVO] was only 60% of that observed for the wild-type PRS[WT]. To gain deeper insights into the enzyme, the three-dimensional structure of the synthetase, along with its substrate ATP, was modeled using AlphaFold2. This *in-silico* study suggested that the hydrogen bonds between N38 and ATP were weaker than those between D38 and ATP (Fig. 2d), implying that PRS[EVO] might have reduced binding affinity for ATP. These findings indicated that the mutations in *prs* resulted in a partial loss-of-function of the enzyme contributing to the growth advantage observed in the SE01.1 strain under methanol limitation.

Given that the evolved SE01.1 strain harbored multiple mutations, we sought to ascertain whether the partial loss of PRS function alone could confer advantageous growth under low methanol concentrations. To test this, the *prs[EVO]* allele was introduced into *prs[WT]* in the wild-type *Mr. extorquens* AM1, resulting in the strain named AM1[PTR]. This strain was cultivated at four low methanol concentrations (15 mM, 10 mM, 5 mM, and 1.2 mM) that mimic the phyllosphere environment[14,15], as well as at 0.5 mM, which was expected to be close to the calcium-dependent methanol dehydrogenase Km(app)[32]. The lag phase of the AM1[PTR] strain was 1–2.5 hours shorter than that of the AM1[WT] strain (Fig. 2e). The maximum OD600 increased by 7.0–26.7% compared to the AM1[WT] strain when grown on methanol concentrations ranging from 0.5 to 15 mM methanol (Fig. 2e). Notably, the growth rate of the AM1[PTR] strain was indistinguishable from that of the AM1[WT] strain on low methanol concentrations, while biomass yields increased by 11.0% and 8.3% on 10 mM and 15 mM methanol, respectively (Fig. 2f). These findings suggested that PRS[EVO] achieved a higher yield without any noticeable decrease in growth rate. The growth fitness of the AM1[PTR] strain followed a similar trend to that of the AM1[WT] strain with 60 mM methanol, albeit with a slightly longer lag phase and

a slower growth rate observed at 120 mM methanol despite reaching the same final biomass level (Supplementary Fig. 8). We hypothesized that PRS[EVO] might result in inadequate synthesis of nucleotides at standard methanol concentrations, consequently impeding genome synthesis and cell replication.

Subsequently, we aimed to investigate the impact of down-regulating *prs* on the growth fitness of *Mr. extorquens* AM1 under low methanol concentrations. To achieve this, we utilized CRISPRi to inhibit transcription initiation of the *prs* gene, following previously described methods[33]. The growth data of the *Mr. extorquens* AM1 strains are illustrated in Fig. 2g. Similar to the AM1[PTR] strain, the culture of *Mr. extorquens* AM1 with tuned-down *prs* expression exhibited significantly improved growth under low methanol concentrations (Fig. 2g, h).

Furthermore, we analyzed the growth of the AM1[PTR] strain and the AM1[WT] strain with C2 (acetate, ethanol, and oxalate), C3 (pyruvate), and C4 (succinate) substrates as the sole carbon sources. All selected C2 to C4 substrates are known to be present in the plant phyllosphere and rhizosphere[2,21,34]. Similar to the methanol results, several growth advantages, including a shorter lag phase and higher biomass, were observed on all carbon sources at 1 mM and 0.1 mM (Fig. 2i and Supplementary Fig. 9). Furthermore, growth advantages of the AM1[PTR] strain were evident when exposed to combinations of limited quantities of all carbon sources from C1 to C4 (Fig. 2j). These observations indicated that reducing PRS activity could enhance the growth of *Mr. extorquens* under various nutrient-limited growth conditions. Consequently, PRS seemed to play a broader metabolic role beyond solely increasing the efficiency of C1-carbon metabolism.

## PRS is a metabolic valve reallocating the carbon flow by up-regulating the phosphoketolase (PKT) pathway

To gain further insights into how PRS[EVO] enhanced the growth on low methanol concentrations, we compared the targeted metabolites and transcripts in the *Mr. extorquens* strains AM1[PTR] and AM1[WT] when cultivated with 10 mM methanol (Fig. 3a, b, Supplementary Data 1 and 2). Supplementary Data 1 and 2 present the metabolites and genes associated with the central mechanism, along with fold-change values and *P* values.

Numerous nucleotides, including ADP, AMP, UDP, IMP, GDP, CMP, and IDP, exhibited significant decreases in pool size in the AM1[PTR]

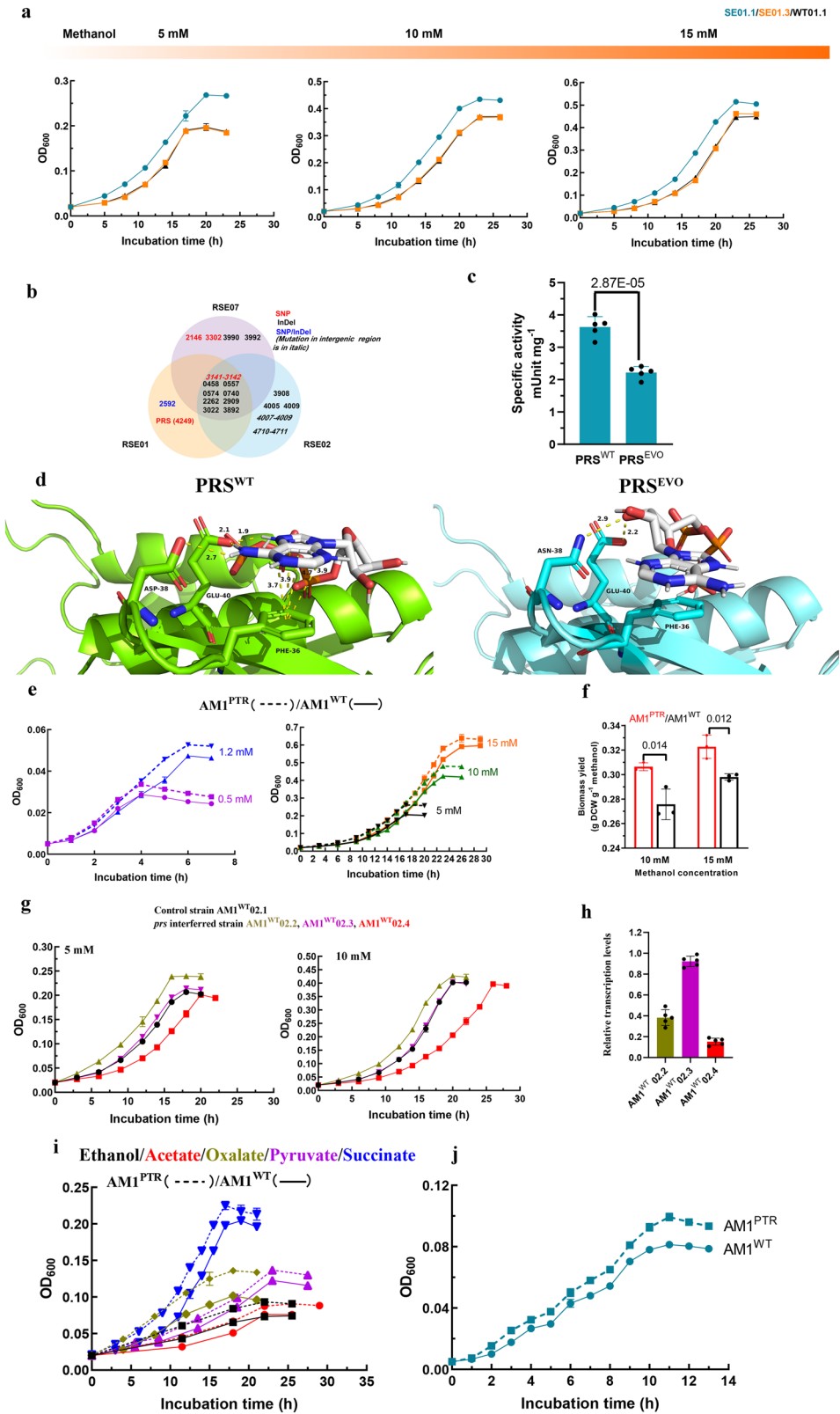

strain, with ADP showing the most substantial decrease of approximately 0.7-fold (Fig. 3a). Histidine and NADP biosynthesis, interconnected with nucleotide synthesis, displayed a similar pattern to the nucleotides (Supplementary Data 1). Fructose-6-phosphate (F6P) was reduced by 0.4-fold, and the pools of fructose-1,6-bisphosphate (FBP), glucose-6-phosphate (G6P), and glucose-1-phosphate (G1P) were 0.4-fold lower compared to the AM1$^{WT}$ strain. Notably, acetyl-phosphate

(acetyl-P), which is readily converted to acetyl-CoA by the PKT pathway, was 2.8-fold higher in the AM1$^{PTR}$ strain (Fig. 3a). Genes $xfp$ ($xfp$1, $xfp$2), $pta$, and $ack$ ($ack$1, $ack$3) involved in the PKT pathway were significantly up-regulated by over 2-fold in the AM1$^{PTR}$ strain compared to the AM1$^{WT}$ strain (Fig. 3b and Supplementary Data 2). The data suggested a depletion of the F6P pool and, consequently, an increase in the acetyl-P pool, likely due to an up-regulated PKT pathway.

**Fig. 2 | Partial loss-of-function of PRS increases the growth of *Mr. extorquens* AM1 on low concentrations of carbon sources. a** The evolved strain SE01.1 showed growth advantages over the control strain AM1$^{WT}$01.1 on low methanol concentrations, while PRS$^{EVO}$ back mutated to PRS$^{WT}$ (SE01.3) eliminated the growth advantage of SE01.1. The strains were initially grown on 120 mM methanol and then transferred to low methanol concentrations of 5 mM, 10 mM, and 15 mM. **b** SNP and indel mutations occurred in the evolved strains RSE01, RSE02, and RSE07. Twenty mutated genes or intergenic regions were identified. **c** Comparison of specific activities of the purified PRS$^{WT}$ and PRS$^{EVO}$. **d** The three-dimensional structure of PRS and its substrate ATP showed weaker binding of ATP by PRS$^{EVO}$. **e** The growth curves of the AM1$^{WT}$ and the AM1$^{PTR}$ strain at five low methanol concentrations (0.5 mM, 1.2 mM, 5 mM, 10 mM, and 15 mM). **f** Biomass yields of the AM1$^{WT}$ and the AM1$^{PTR}$ strain grown on methanol at 10 mM and 15 mM.

DCW was abbreviated for "dry cell weight". **g** The growth curves of the *prs* inter-fered strains and the control strain (carrying the CRISPRi plasmid without small guide RNA) at 5 mM and 10 mM methanol. **h** The relative abundance of *prs* transcription in the interfered strains at 10 mM methanol. **i** The growth curves of the AM1$^{WT}$ and the AM1$^{PTR}$ strains grown on multi-carbon sources at 1 mM. **j** The growth curves of the AM1$^{WT}$ and the AM1$^{PTR}$ strains on a combination of methanol at 0.5 mM and multi-carbon sources (containing 0.1 mM ethanol, 0.1 mM acetate, 0.1 mM oxalate, 0.1 mM pyruvate, and 0.1 mM succinate). Data are presented as mean ± SD (*n* = 6 biologically independent samples with three technical repeats for (**a**, **e**, **g**, **i** and **j**); *n* = 5 biologically independent samples for (**c**, **h**); *n* = 3 biologically independent samples for (**f**). Significance was analyzed using a two-tailed t-test analysis. Source data are provided as a Source Data file.

Moreover, changes in the central carbon metabolism involved in the methanol assimilation pathway, specifically the serine cycle and the ethylmalonyl-CoA (EMC) pathway, were also explored. The 3-hydroxybutyl-CoA pool (measured as its acid derivative) and poly-hydroxybutyrate (PHB) exhibited a trend of being higher in the AM1$^{PTR}$ strain (Fig. 3a). The pool sizes of other EMC pathway intermediates, such as ethylmalonyl-CoA, methylsuccinyl-CoA, methylmalonyl-CoA (measured as its acid derivatives), and the serine cycle intermediates glycerate, 2-phosphoglycerate, phosphoenolpyruvate, and malate, did not change significantly (Fig. 3a and Supplementary Data 1). Genes associated with the EMC pathway, including *phaA*, *phaB*, *phaC*, *mcmAB*, and *mcl*, as well as genes involved in the H$_4$F-dependent formate transfer pathway and the serine cycle (*ftfl*, *fch*, *ppc* and *mdh*) and genes in the gluconeogenic pathway (*gapA*, *pgk*, and *cbbA*) exhibited the expression changes ranging from 2.3- to 7.1-fold when comparing the AM1$^{PTR}$ strain and AM1$^{WT}$ strain (Fig. 3b). The genes involved in methanol oxidation, including *mxaFJG*, *fae*, *mtdB*, *mch*, and *fhcABCD* did not show significant changes (either absolute log$_2$-fold change less than one or *p*-value over 0.05). Notably, the transcription of *fdh1234*, encoding formate dehydrogenase (FDH) that catalyzes the major reaction for CO$_2$ loss[35], was found to be down-regulated by 0.3- to 0.7-fold in the AM1$^{PTR}$ strain, and accordingly, the FDH activity was decreased by 17% in the AM1$^{PTR}$ strain (Fig. 3b, c). These declines appeared to coincide with a formate branchpoint redistribution that complemented the enhanced methanol assimilation.

Considering that the PKT pathway played a non-essential role in the growth of *Mr. extorquens* on methanol[26] (Supplementary Fig. 10), we proposed that the up-regulation of the PKT pathway indeed laid the foundation for enhancing growth fitness under limited methanol conditions. The *xfp* gene plays a pivotal role in the PKT pathway by catalyzing the conversion of F6P or xylulose-5-phosphate (Xu5P) into erythrose-4-phosphate or glyceraldehyde-3-phosphate and acetyl-P[36]. In contrast to the AM1$^{WT}$ strain, where the deletion of *xfp*2 did not alter the growth trend at low methanol concentrations, the deletion of *xfp*2 expression in the AM1$^{PTR}$ strain (AM1$^{PTR}$04) notably reduced growth advantages (Fig. 3d). Furthermore, interference with *xfp*2 in the AM1$^{PTR}$ strain resulted in decreased growth at low methanol concentrations (Supplementary Fig. 11). Taken together, these findings demonstrated that PRS served as a metabolic valve that redirected carbon flow from nucleotide synthesis towards the central assimilation through the up-regulated PKT pathway (Fig. 3e). As a result, the mutant strain AM1$^{PTR}$ efficiently assimilated limited methanol while experiencing growth fitness benefits. The mechanisms that underlie the enhanced utilization of limited multi-carbon sources, as well as combinations of all carbon sources from C1 to C4, by the PRS mutation, are currently under investigation and will be reported in a subsequent publication.

## Partially turning off the PRS valve benefits growth in *Methylobacterium*/*Methylorubrum*

To evaluate the significance of the observed phenotypes within other *Methylobacterium*/*Methylorubrum* species, we conducted an investigation into the distribution of PRS and sequence homologies among 62 representatives of *Methylobacterium*/*Methylorubrum*[37]. Our findings revealed that the amino acid residue at position 38 was conserved across all *Methylobacterium*/*Methylorubrum* species (Fig. 4a). Furthermore, we identified genes associated with the PKT pathway and EMC pathway in 35 out of 62 *Methylobacterium*/*Methylorubrum* species (Supplementary Fig. 12).

Given the metabolic step's conservation among *Methylobacterium*/*Methylorubrum* species, we hypothesized that the observed growth benefit at low methanol concentrations might be a common trait in these bacteria. To test this hypothesis, we introduced the *prs*$^{EVO}$ allele into a representative epiphyte of leaf surfaces of *Arabidopsis*, *Mr. extorquens* PA1[38], and three well-known plant growth-promoting *Methylobacterium* species, *Mb. nodulans* ORS2060[39], *Mb. radiotolerans* JCM2831[39], and *Mb. oryzae* CBM20[40]. The latter three species were chosen for their evolutionary distinct from *Mr. extorquens* (Supplementary Fig. 12). The mutated strains, namely named PA1$^{PTR}$, ND$^{PTR}$, RA$^{PTR}$, and named OR$^{PTR}$, all exhibited a growth advantage under low methanol conditions (Fig. 4b).

Subsequently, we disrupted the transcriptional initiation of *prs* in *Mb. oryzae* CBM20 and *Mb. radiotolerans* JCM2831. In comparison to their control strains, the interfered strains (OR02, OR03, OR04, and RA02, RA03) showed significant growth at a low methanol concentration of 5 mM (Fig. 4c). The growth rate of the interfered *Mb. oryzae* CBM20 (OR02 and OR03) was even 20% higher than that of the control strain (OR01). These results indicated that partially turning off the PRS valve benefiting growth was applicable to a wide range of *Methylobacterium*/*Methylorubrum* species.

## PRS mutation is advantageous in the colonization of *Arabidopsis* leaves

Bacterial growth in vitro can significantly differ from in-situ conditions. To assess the advantages of the partial loss-of-function in PRS for promoting colonization in phyllosphere niches, we conducted a series of colonization experiments. *A. thaliana* seeds were inoculated with either the AM1$^{PTR}$ strain, the PA1$^{PTR}$ strain, the RA$^{PTR}$ strain, or their respective WT strains (approximately 10$^6$ colony-forming units (CFU)) under a gnotobiotic plant growth system (see Methods). At 21 days post-inoculation, the AM1$^{PTR}$ strain reached 2.27 × 10$^8$ CFU per gram of leaf fresh weight (CFU g$^{-1}$ FW), while the AM1$^{WT}$ strain reached 1.70 × 10$^8$ CFU g$^{-1}$ FW (Fig. 5a). The population sizes indicated that the colonization ability of the AM1$^{PTR}$ strain was 34% higher compared to the AM1$^{WT}$ strain. No trace of mutant or AM1$^{WT}$ strain colonization was detected on control plants (seeds dipped into a 10 mM MgCl$_2$ solution). Similarly, the colonized populations of the PA1$^{PTR}$ strain and RA$^{PTR}$ strain under the same controlled system increased by 59% and 62%, respectively, compared to their WT strains (Fig. 5b, c).

Subsequently, we investigated whether the PRS mutant strain maintained phyllosphere colonization efficiency under competitive conditions. The AM1$^{PTR}$ and AM1$^{WT}$ strains were pitted against a white *Mr. extorquens* AM1 strain (YAIP strain) created by deleting the *crtI* gene

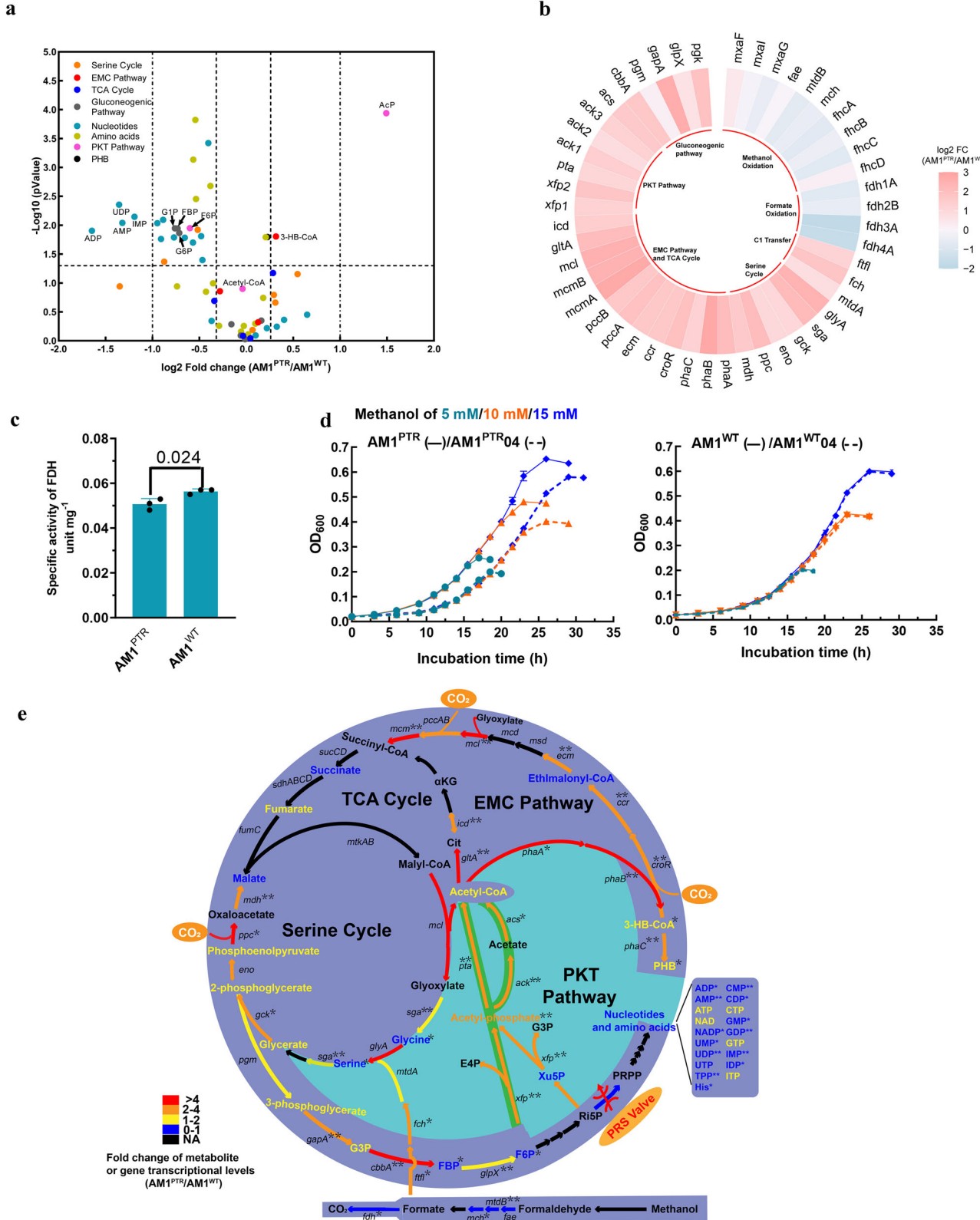

(META1_3655), which is involved in carotenoid synthesis[41]. This mutation does not impede the growth of *Mr. extorquens* under standard cultivation conditions[33,41]; however, it caused slight growth impairment when colonizing plants (Fig. 5d), as carotenoid is considered crucial under light-conditions and thus for plant colonization[42]. The direct color of the colonies served as a measure of competitive fitness. The AM1^PTR strain appeared to exhibit faster growth than its relatively poor-

colonizing counterparts and reached $1.5 \times 10^8$ CFU g$^{-1}$ FW at 21 days, marking an 88% increase compared to the YAIP strain, while the AM1^WT strain exhibited competitive performance similar to the YAIP strain (Fig. 5d). The stability in colonization efficiency under a competitive condition involving two strains suggested that the presence of other *Methylorubrum/Methylobacterium* strains likely did not compromise the performance of the PRS mutant strain.

**Fig. 3 | PRS, as a metabolic valve, reallocates carbon flows through nucleotide anabolism and methylotrophic assimilation by up-regulating the PKT pathway. a** Comparison of the intermediates involved in the central methylotrophic metabolism and nucleotide synthesis of the AM1$^{PTR}$ and AM1$^{WT}$ strains grown on 10 mM methanol ($n = 3$ biologically independent samples.). **b** Transcriptional comparison of gene expression between the AM1$^{PTR}$ strain and AM1$^{WT}$ strain grown on 10 mM methanol ($n = 3$ biologically independent samples with five technical repeats). **c** Specific activities of FDH in the strains of AM1$^{PTR}$ and AM1$^{WT}$ grown on 10 mM methanol ($n = 3$ biologically independent samples with three technical repeats). **d** Knocking out the gene *xfp*2 (AM1$^{PTR}$ 04) decreased the growth advantage of the AM1$^{PTR}$ strain on low methanol concentrations of 5 mM, 10 mM, and 15 mM, while it did not affect the growth of the AM1$^{WT}$ strain ($n = 5$ biologically independent samples with three technical repeats.). **e** Schematic central

methylotrophic network in the AM1$^{PTR}$ strain on low methanol concentration. Partially turning off the PRS valve decreased the allocation of limited methanol for nucleotide synthesis and up-regulated the PKT pathway to channel more methanol through the assimilation pathways. Reversible reactions are shown as double arrows. Fold changes of gene transcriptional levels in the AM1$^{PTR}$ strain versus the AM1$^{WT}$ are indicated by colored arrows. Fold changes of intermediate pools in the AM1$^{PTR}$ strain versus the AM1$^{WT}$ are indicated by colored metabolites. "NA" is abbreviated for "not analyzed". F6P fructose-6-phosphate, G6P glucose-6-phosphate, G1P glucose-1-phosphate, Pyr pyruvate, FBP fructose-1,6-biphosphate, 3-HB-CoA 3-hydroxybutyryl-CoA, AcP acetyl-phosphate, PHB poly-β-hydroxybutyrate. Data are presented as mean ± SD. Significance was analyzed using a two-tailed t-test analysis. Source data are provided as a Source Data file.

We also quantified the dynamics of phyllosphere abundance of the AM1$^{PTR}$ and AM1$^{WT}$ strains on *A. thaliana* under growth chamber conditions to assess the colonization efficiency of the AM1$^{PTR}$ strain in non-sterile conditions. *A. thaliana* seeds were planted in non-sterile soil and watered with tap water. After growing for 7 days, *A. thaliana* seedlings were foliar sprayed with either the AM1$^{PTR}$ strain or AM1$^{WT}$ strain. The control group was foliar sprayed with 10 mM MgCl$_2$. To determine the abundance of the AM1$^{PTR}$ or AM1$^{WT}$ strain in the phyllosphere, we designed specific genomic region primers for qPCR analysis of *Mr. extorquens*. The AM1$^{PTR}$ strain was 2.5-fold more abundant than the AM1$^{WT}$ strain in the phyllosphere at any given time point (Fig. 5e). Trace abundances of both AM1 strains were detectable in control plants, most likely due to a fresh air system flow, and were assumed to be at the background level for the growth chamber experimental setup.

Finally, we investigated whether the PKT pathway alone could serve as an essential function for enhancing phyllosphere colonization. The AM1$^{PTR}$04 strain (AM1$^{PTR}$Δ*xfp2*) lost the ability to efficiently colonize the *Arabidopsis* phyllosphere, while the control strain AM1$^{WT}$04 (AM1$^{WT}$Δ*xfp2*) had no impact on plant colonization (Fig. 5f). Plant leaves can release various types of multi-carbon sources, such as oxalate and succinate. The central metabolic pathway for utilizing multi-carbon sources by *Methylobacterium/Methylorubrum* strains differs from methylotrophy[43,44]. The colonization of the AM1$^{PTR}$04 strain suggested that the activation of the PKT pathway might also influence the efficiency of *Methylobacterium/Methylorubrum* strains in utilizing these multi-carbon sources in the phyllosphere. In summary, these results indicated that partially turning off PRS coupling to the PKT pathway effectively promoted the colonization of *Methylobacterium/Methylorubrum* strains in the phyllosphere.

**The PRS mutant strain improves plant growth and yield with spraying at low abundance**

In agriculturally relevant plant growth conditions, both biotic and abiotic factors can diminish the survival and colonization of microbial inoculants. Our initial focus was on determining whether the efficient colonization of the PRS mutant strain could enhance plant growth in growth chambers, even with low-abundance foliar spraying, a crucial consideration for achieving anticipated results in agricultural trials. We applied the RA$^{PTR}$ strain or the RA$^{WT}$ strain by spraying a low abundance of one-milliliter culture ($4 \times 10^4$ CFU/mL) per plant on *Brassica chinensis* (Pak Choi) of the Brassicaceae family, and *Cucumis melo* and *Cucumis sativus* of the Cucurbitaceae family in a non-sterile growth chamber (Fig. 6a). The average fresh weight per plant of *B. chinensis* seedlings (4.14 g/plant) sprayed with the RA$^{PTR}$ strain on the 24th day of growth was 36% higher than that of the RA$^{WT}$ strain (4.14 g/plant compared to 3.04 g/plant), and 30% higher than that of the control group treated with 10 mM MgCl$_2$ (3.19 g/plant) (Fig. 6b, c). Similarly, the average fresh weights per plant of *C. melo* and *C. sativus* seedlings sprayed with a low abundance of the RA$^{PTR}$ strain were 42% and 18% higher than those of the RA$^{WT}$ strain on the 24th day of plant growth,

respectively (Fig. 6b, c). Consistently, the number of colonized cells per gram of leaf fresh weight of the RA$^{PTR}$ strain on *B. chinensis, C. melo*, and *C. sativus* at both the 7th and 12th day post-inoculation was significantly higher than that of the RA$^{WT}$ strain (Fig. 6d). The drastic decrease in colonized cells per gram of leaf fresh weight at the 12th day post-inoculation was likely due to the rapid extension of plant leaves. The PRS mutation in the RA strain did not seem to affect the synthesis of indoleacetic acid and ACC deaminases under laboratory cultivation conditions (Supplementary Fig. 13). Therefore, we hypothesized that the higher plant growth-promoting ability of the RA$^{PTR}$ strain might be attributed to greater number of strains colonizing the phyllosphere.

*B. chinensis*, a widely cultivated leafy vegetable in China, Asia, Europe, and America, serves as a crucial component in local diets. To assess whether the PRS mutant strain could enhance the yield of *B. chinensis* under agricultural management, we cultivated *B. chinensis* in a standard Chinese agricultural solar greenhouse with a temperature range of 10 °C to 20 °C and a humidity range of 50% to 60% (as described in Methods). The RA$^{PTR}$ or RA$^{WT}$ strain was applied by spraying at a low abundance similar to that in the growth chamber. *B. chinensis* was harvested on the 40th day of growth, reaching an optimal size for the market. Consistent with the results observed in the growth chamber, the number of colonized cells per gram of leaf fresh weight of the RA$^{PTR}$ strain on *B. chinensis* phyllosphere was more than two-fold higher than that of the RA$^{WT}$ strain throughout the growth period (Fig. 6e). The average fresh weight per hundred plants sprayed with the RA$^{PTR}$ strain (1.82 kg/hundred plants) was 24% higher than that of the RA$^{WT}$ strain (1.47 kg/hundred plants) and 18% higher than that of the control group (1.54 kg/hundred plants) treated with 10 mM MgCl$_2$ (Fig. 6f, g). These findings suggested that the RA$^{PTR}$ strain significantly contributed to enhanced *B. chinensis* yield under agricultural conditions.

## Discussion

In this study, we identified the pivotal role of PRS as a metabolic valve, influencing trade-offs between nucleotide anabolism and central assimilatory metabolism in *Methylobacterium/Methylorubrum* (Fig. 1). These metabolic traits not only enhanced growth under conditions of limited methanol availability but also facilitated phyllosphere colonization. Moreover, these traits also proved beneficial for plant growth and yield, even when subjected to low-abundance *Methylobacterium/Methylorubrum* spraying. Importantly, the function of PRS was not an isolated phenomenon but was prevalent among *Methylobacterium/Methylorubrum* strains. The PRS valve introduced a strategy for facilitating the establishment of beneficial traits in *Methylobacterium/Methylorubrum*, positively impacting plant growth and yield.

Microbes typically uphold a trade-off between catabolism and anabolism in nutrient-scarce environments, leading to decreased growth[45]. Our primary objective focused on enhancing phyllosphere colonization by improving central metabolic efficiency under conditions of limited methanol availability. A crucial strategy in this

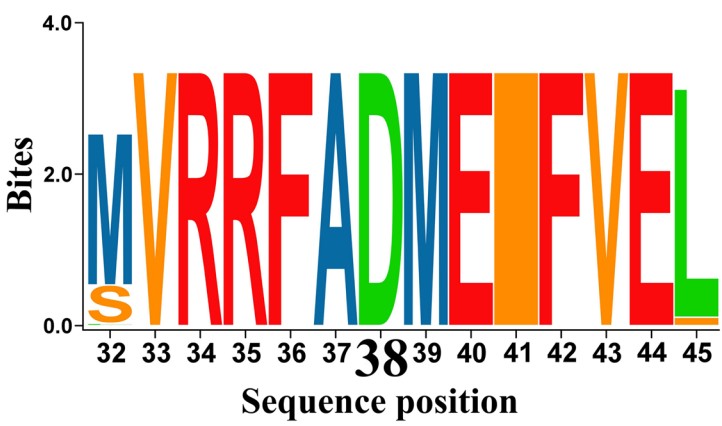

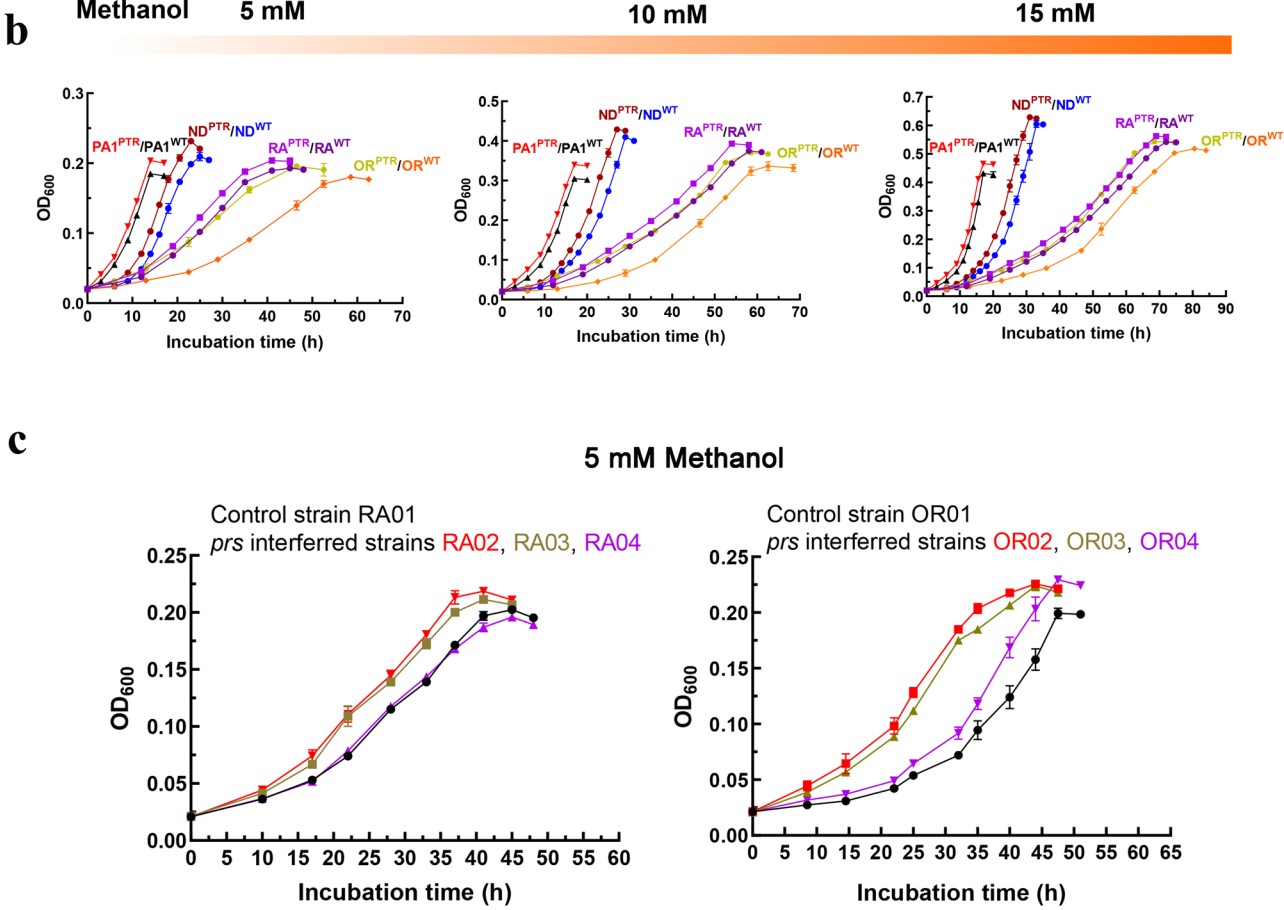

**Fig. 4 | Partially turning off PRS increases the growth of *Methylobacterium/Methylorubrum* species on low methanol concentrations. a** SeqLogo schematic showing the conservation of amino acid residues of PRS in all *Methylobacterium/Methylorubrum* species. **b** Comparison of the growth curves between the ND$^{WT}$ and ND$^{PTR}$ strains, between the RA$^{WT}$ and RA$^{PTR}$ strains, between the PA1$^{WT}$ and PA1$^{PTR}$ strains, and between the OR$^{WT}$ and OR$^{PTR}$ strains on low methanol concentrations of 5 mM, 10 mM, and 15 mM ($n = 6$ biologically independent samples with three technical repeats). **c** Interfering *prs* in *Mb. radiotolerans* JCM2831 and in *Mb. oryzae* CBM20 improved the growth on 5 mM methanol ($n = 6$ biologically independent samples with three technical repeats). Data are presented as mean ± SD. Source data are provided as a Source Data file.

endeavor involved optimizing nucleotide anabolism, a process known for its high energy demand, reliant upon H$_4$F derivatives and glycine[46]. Notably, both of these compounds also serve as intermediates within the serine assimilation cycle[47]. Partial impairment of PRS function led to a more prudent allocation of limited methanol resources, redirecting them away from nucleotide synthesis (Fig. 3). This adaptation effectively alleviated direct competition with core methylotrophic assimilation pathways. It is notable that previous investigations have

documented instances of PRS mutations emerging during the evolutionary history of autotrophic *E. coli* and methylotrophic *E. coli*[48,49]. These mutations have been observed to reduce the branching of intermediates, such as Ri5P, away from the constructed Calvin-Benson-Bassham cycle or the RuMP cycle, which is essential for maintaining a steady precursor supply to support autocatalytic cycles[50]. Interestingly, our findings in *Mr. extorquens* AM1 revealed a distinct role for the PRS mutation. Instead of stabilizing the serine cycle, it optimized the

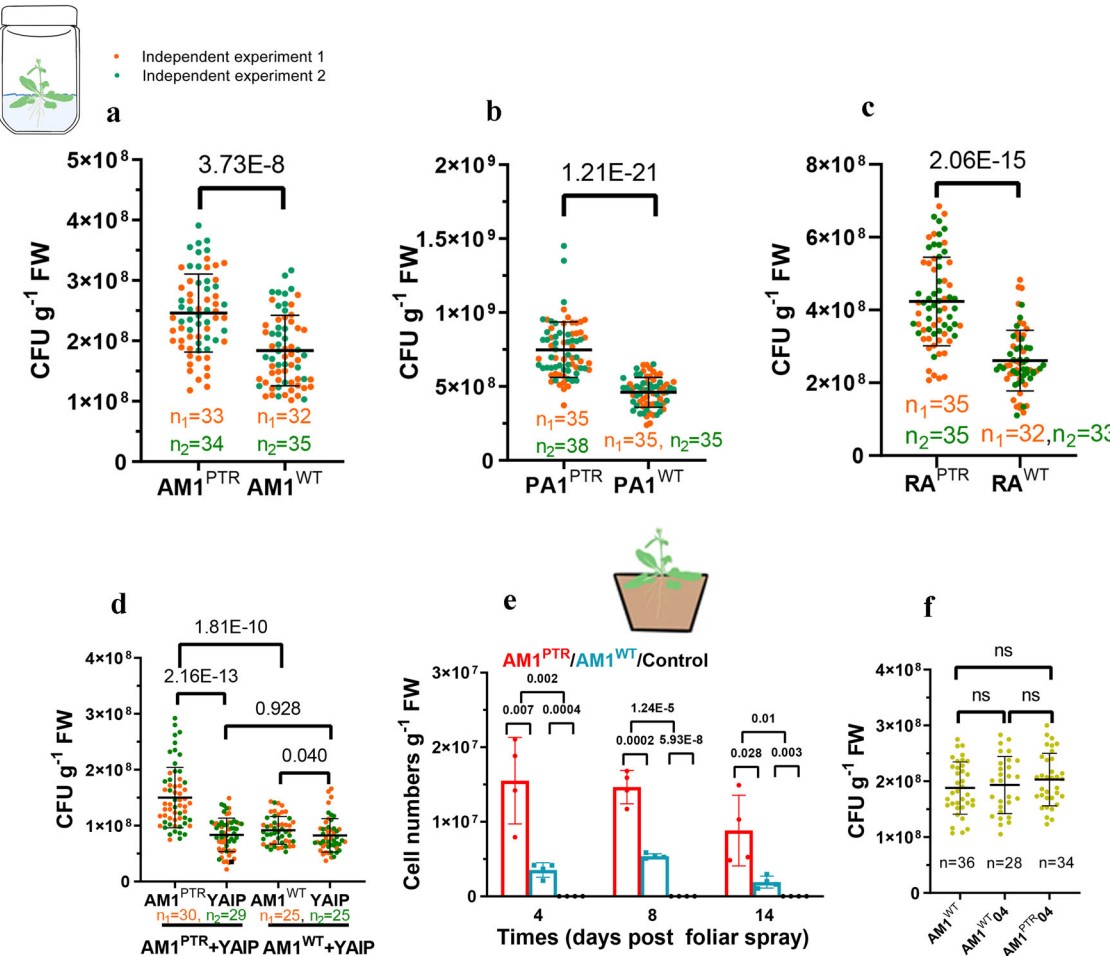

**Fig. 5 | PRS mutation promotes *Methylobacterium/Methylorubrum* colonization on *Arabidopsis* leaves. a–c** Comparison of the population size of the strains of AM1^PTR and AM1^WT, PA1^PTR and PA1^WT, as well as RA^PTR and RA^WT on *Arabidopsis* leaves using a gnotobiotic growth system. **d** The AM1^PTR strain showed higher colonization efficiency under two strain-competitive condition. The AM1^PTR and AM1^WT strains were each inoculated with the YAIP strain on *Arabidopsis* seeds in a gnotobiotic growth system. **e** Comparison of the population size of the AM1^PTR and AM1^WT strains on *Arabidopsis* leaves at days 4, 8, and 14 post foliar spray in the growth chamber (*n* = 4 independent biological subgroups.). Significance was analyzed using one-way ANOVA with posthoc tests. **f** Knocking out of *xfp*2 in the AM1^PTR strain (AM1^PTR04) eliminated advanced colonization on *Arabidopsis* leaves compared to the AM1^WT04 strain (*Mr. extorquens* AM1Δ*xfp2*) and the AM1^WT strain. For (**a–d, f**), significance was analyzed using a two-tailed t-test analysis with log-transformation. ns means no significance. Data are presented as mean ± SD. Source data are provided as a Source Data file.

likelihood of generating acetyl-CoA through the up-regulation of the originally non-essential PKT pathway.

The carbon-neutral PKT pathway, responsible for acetyl-CoA production, is a characteristic found in diverse microbial taxa, encompassing bifidobacteria and cyanobacteria[36,51]. Genes associated with this pathway have also been identified within the alphaproteo-bacterial methylotrophs[52]. However, in the case of *Mr. extorquens* AM1, the substrates required for the PKT pathway, namely F6P and Xu5P, require generation via a reverse glycolytic pathway, which involves the expenditure of both ATP and NADH. Acetyl-CoA is primarily synthe-sized through the degradation of malyl-CoA within the serine cycle, ensuring minimal carbon loss[29,47]. Consequently, under standard cul-tivation conditions, the contribution of the PKT pathway is considered negligible[26]. However, under conditions of limited methanol avail-ability, *Mr. extorquens* AM1 appeared to produce an excess of unne-cessary nucleotides for DNA and RNA synthesis, resulting in carbon wastage and suboptimal methanol utilization. By partially reducing PRS activity, we effectively attenuated the diversion of F6P and Xu5P into PRPP biosynthesis while concurrently enhancing the PKT pathway (Fig. 3). Subsequently, the EMC pathway assimilated acetyl-CoA pro-duced through the PKT pathway, facilitating increased $CO_2$ fixation

and glyoxylate production. Glyoxylate was then condensed with 5,10-methylene-$H_4F$ to expedite methanol assimilation via the serine cycle. Concomitantly, the up-regulation of central assimilation was accom-panied by the down-regulation of complete formate oxidation. A recent study has also observed that methanotrophic strains growing under low methane conditions tend to reduce carbon loss as formate or $CO_2$ to facilitate increased carbon assimilation[53]. The reallocation of resources towards fortifying central methanol assimilation, at the expense of nucleotide anabolism, likely acquired the necessary com-ponents to achieve this metabolic shift. Consequently, this metabolic reprogramming provided a cost-effective and logical exchange, effi-ciently utilizing limited carbon, energy, and reducing equivalents for both nucleotide anabolism and the circulation of central metabolism. This led to an increase in biomass yields and conferred enhanced competitiveness in terms of growth and the colonization abilities of plants (Figs. 2, 4, and 5).

According to ecological theory, in situations where nutrient resources are scarce and must be distributed among a larger number of organisms, the average reproductive success per cell tends to decrease[54]. Within the phyllosphere, the local "carrying capacity" for bacteria typically spans from $10^6$ to $10^8$ cells per square centimeter of

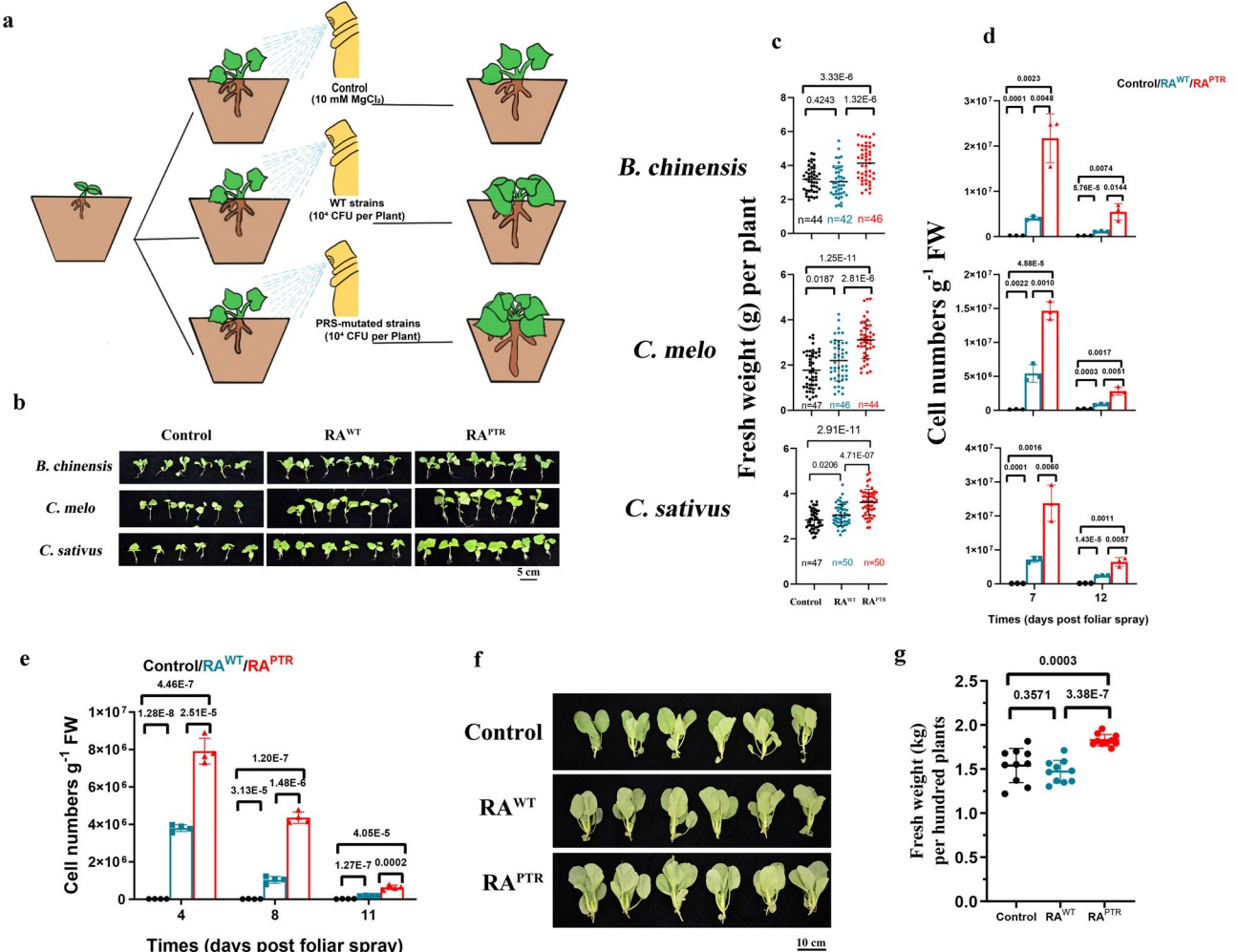

**Fig. 6 | The PRS mutant strain enhances seedling growth and plant yield following foliar spraying with a low abundant dose. a** Illustration of the spraying of plants with either WT *Methylobacterium/Methylorubrum* or PRS mutant *Methylobacterium/Methylorubrum* with a low abundant dose. **b, c** The RA$^{PTR}$ strain effectively improved the growth of *B. chinensis, C. sativus,* and *C. melo* seedlings in the growth chamber. Pictures and data were taken at 12 days post foliar spray (plant age of 24-day). **d** Spraying a low dose of the RA$^{PTR}$ strain improved the growth of the plant seedlings due to increased colonization capacity (*n* = 3 independent biological subgroups). **e, f** The RA$^{PTR}$ strain increased the yield of *B. chinensis* in agricultural solar greenhouses. Pictures and data were taken when *B. chinensis* were 40 days old (*n* = 4 independent biological subgroups). **g** Spraying a low dose of the RA$^{PTR}$ strain improved the yield of *B. chinensis* due to increased colonization capacity (*n* = 10 independent biological subgroups). For (**c, f**), significance was analyzed using a two-tailed t-test analysis; for (**d, g**), significance was analyzed using one-way ANOVA with posthoc tests. Data are presented as mean ± SD. Source data are provided as a Source Data file.

leaf tissue[55], which is notably lower than what is observed in the rhizosphere. The phyllosphere presents niches with limited and fluctuating carbon availability, reducing the likelihood of microbial reproduction and the colonization of plants by microbial supplements. This phenomenon may elucidate significant disparities observed for the growth-promoting microbes in laboratory settings versus in their natural habitats[56]. Certain strains of *Methylobacterium/Methylorubrum* exhibit a remarkable capacity to exploit underutilized organic carbon reservoirs and utilize sunlight for carbon assimilation and energy conservation within the phyllosphere[21,57]. Conversely, some strains have evolved into auxotrophic variants through the acquisition of leaf-derived compounds, thereby economizing on precursors and reducing energy expenditures for vital biological processes[58]. Recent observations concerning priority effects suggest that the early introduction of phyllosphere-associated microbes during microbial community establishment may enhance colonization success[59]. In this regard, the application of a low abundance of *Mb. radiotolerans* JCM2831 with a single-nucleotide mutation of PRS significantly enhanced the growth of plant seedlings in growth chambers and the yield of *B. chinensis* in agricultural solar greenhouses (Fig. 6). This

metabolic characteristic held substantial promise for agricultural utilization. Prospective developments in this area might include the implementation of targeted enrichment studies and ALE studies to isolate or select traits with similar characteristics. Such endeavors could power probiotic *Methylobacterium/Methylorubrum* strains, which inherently exhibited limited population assembly on certain plant species, to more efficiently stimulate the growth and yield of these plants. Given the demonstrated dose-dependent response in the interplay between plants and microbial inoculants[60], agricultural production expenses could be curtailed through the application of a reduced quantity of the designated *Methylobacterium/Methylorubrum* inoculant once substantial colonization is achieved.

To summarize, the partial inhibition of PRS in *Methylobacterium/Methylorubrum* resulted in a reduction in the allocation of limited methanol resources towards nucleotide anabolism. This, in turn, led to an up-regulation of the PKT pathway, facilitating increased carbon flow through central carbon assimilation. This metabolic adjustment ultimately enhanced phyllosphere colonization and contributed to the promotion of plant growth and yield. It is crucial to note that the broader applicability of this discovery in agricultural management

should be assessed in future studies. Factors such as plant host species, variations in the phyllosphere environment, microbial community composition, and agricultural practices like fertilizer application may all impact colonization efficacy[6,61]. Integrating laboratory findings with field-scale trials holds substantial promise for refining *Methylobacterium*/*Methylorubrum* traits as optimal plant probiotics for future agricultural applications.

## Methods

### Bacteria, media, and culture conditions

All bacterial strains utilized in this study are listed in Supplementary Table 1, and detailed information regarding the plasmids used can be found in Supplementary Table 3. The *Mr. extorquens* strains were routinely maintained on a mineral salt medium (MSM)[62] supplemented with methanol as the sole carbon and energy source unless otherwise specified. The concentration of methanol varied, ranging from 120 mM to lower concentrations (0.5 mM, 1.2 mM, 5 mM, 10 mM, and 15 mM). *Mr. extorquens* strains were cultivated in tubes at 30 °C until reaching the mid-exponential phase, after which they were sub-cultured into 50 mL of MSM in 250-mL flasks with an initial $OD_{600}$ of 0.02 or 0.005. These flasks were then incubated at 30 °C on a rotary shaker at 200 rpm. The *Mb. nodulans* ORS2060, *Mb. oryzae* CBM20 and *Mb. radiotolerans* JCM2831 strains were cultured similarly to *Mr. extorquens* strains. The *E. coli* BL21 (DE3) strains were cultured in Luria-Bertani (LB) medium at 37 °C. Ampicillin, tetracycline, and rifamycin were used at concentrations of 50 µg/mL, 20 µg/mL, and 50 µg/mL, respectively. All chemical agents were purchased from Sigma-Aldrich (St. Louis, MO, USA).

### ALE

An STSP evolutionary experiment was employed to evolve the ancestor RS03 strain[29]. Seven independent lineages of the RS03 strains were inoculated into 50 mL MSM with 5 mM succinate and 100 mM methanol as the combined carbon sources. The optical density ($OD_{600}$) was measured every 2 days after the consumption of succinate. When visible growth of the cells was observed on the 16th and 25th days, the cultures were diluted and plated on MSM agar plates with 120 mM methanol. The colonies from the plates were then transferred into a liquid medium containing 120 mM methanol to isolate the strains growing on methanol. The stably growing strains were stored for further experimental analysis.

### Growth measurement of *Methylobacterium*/*Methylorubrum* strains

To evaluate growth ability in liquid medium, the *Mr. extorquens* strains were transferred from MSM with 120 mM methanol to 250-mL flasks containing 50 mL of MSM with methanol at varying concentrations or with alternative carbon sources as indicated. The initial $OD_{600}$ was adjusted to 0.02 or 0.005, and the strains were then incubated on a rotary shaker at 30 °C to monitor their growth curves. The $OD_{600}$ of the cultures was measured using a spectrophotometer (GENESYS 10S Vis, Thermo Fisher Scientific, MA, USA) with a 0.5 mL aliquot. CFU was counted at different incubation times to determine growth curves for methanol concentrations of 0.5 mM and 1.2 mM, as well as for multi-carbon sources at 0.1 mM. To quantify the number of CFUs, the culture broth underwent serial dilution. Specifically, 100 µL of the culture broth was mixed with 900 µL of MSM. Subsequently, 100 µL of this mixture was combined with 900 µL of fresh MSM. This process was repeated 2–6 times. Finally, 100 µL of the mixture with various dilutions was spread onto MSM agar plates with methanol at different concentrations as specified. The MSM agar plates were then incubated at 30 °C. The number of colonies on plates containing 30–300 colonies was counted, and the number of CFUs per milliliter was calculated by multiplying by the dilution factor. The period during which the number of cells doubled from the initial inoculation was defined as the lag phase. We monitored the maximum $OD_{600}$ and maximum CFU during the early stationary phase. The specific growth rate was determined by fitting an exponential growth model using Curve Fitter software[63]. Biomass yield on methanol, defined as the amount of cell dry weight generated per gram of methanol consumed, was obtained from linear regression between biomass concentration (g) and substrate methanol concentration (g) during the exponential phase. The dry cell weight was assessed using the following procedure: 50 mL of the culture was extracted and subjected to centrifugation at $8000 \times g$ for 20 min at 4 °C. Following centrifugation, the liquid supernatant was decanted, and the cellular sediment was desiccated at 60 °C for approximately 72 h until a constant weight was achieved, which was considered as the dry cell weight. To determine the number of viable cells corresponding to the maximum $OD_{600}$, colony enumeration was performed on MSM agar plates supplemented with 120 mL methanol after 7 days of incubation at 30 °C. To assess the growth ability on agar solid medium, cultures of *Mr. extorquens* strains were grown at low methanol concentrations of 5 mM, 10 mM, and 15 mM at the exponential phase, and their $OD_{600}$ was adjusted to 0.5. Subsequently, 100 µL of the diluted culture was spread on MSM agar plates containing 5 mM, 10 mM, and 15 mM methanol, respectively. After an incubation period of 3 days at 30 °C, colony counts were conducted. The growth of other *Methylobacterium*/*Methylorubrum* species was evaluated using the same methodology.

### Construction of plasmids and mutant strains

All primers utilized in this study are provided in Supplementary Data 3. Genes were amplified via PCR using PrimeSTAR HS DNA Polymerase and ligated using ClonExpress Ultra One Step Cloning Kit (Vazyme Biotech CO., Ltd., Nanjing, China). Markerless chromosomal gene deletions or allele replacements were performed using the allelic exchange plasmid pCM433[64]. One-kilobase (kb) fragments upstream and downstream of target genes or sites were amplified by PCR, inserted into the pCM433 plasmid, and introduced into the strains via electroporation (Eporator, Eppendorf, Germany). Single-crossover mutant strains were initially selected for tetracycline resistance, and double-crossover mutant strains were subsequently selected by growth on plates containing 5% sucrose (w/v). The mutant strains with successful allele swapping were confirmed using diagnostic PCR and validated by sequencing.

Plasmid pCM80 was employed for gene overexpression in *Mr. extorquens*[65]. The gene *hprA* (META1_1727) was amplified from the genomic DNA of *Mr. extorquens* using the primer pair *hprA*-F and *hprA*-R. The PCR product was inserted into pCM80 under the control of the strong constitutive promoter *PmxaF*. The resulting plasmid was then transformed into the *Mr. extorquens* strains via electroporation, generating the strains AM1^WT01.1, SE01.1, SE02.1, and SE07.1. CRISPRi experiments were conducted following previously described methods[33]. Briefly, to interfere the genes *prs* or *xfp* in the *Methylobacterium*/*Methylorubrum* strains, at least three single guide RNAs (sgRNAs) targeting different regions of the non-template strand of the coding sequence were designed for each gene. Interference plasmids were constructed through reverse PCR using the plasmid pAIO as a template and subsequently introduced into the respective *Methylobacterium*/*Methylorubrum* strains.

### Genome sequencing analysis

The evolved strains were subjected to genome re-sequencing analysis. The genomic DNA of the evolved strains was extracted using the SDS method[66] and sequenced using both the PacBio Sequel platform and Illumina NovaSeq PE150 at Beijing Novogene Bioinformatics Technology Co. For the Illumina NovaSeq platform, the DNA sample was fragmented by sonication to a size of 350 bp, and then the DNA fragments were end-polished, A-tailed, and ligated with the full-length adapter for Illumina sequencing, followed by further PCR amplification. Regarding the PacBio Sequel platform, libraries for single-

molecule real-time sequencing were constructed with an insert size of 20 kb using the single-molecule real-time bell template kit v.1.0. The process involved fragmenting and concentrating DNA, repairing DNA damage and ends, preparing blunt ligation reactions, purifying SMRTbell Templates with 0.45X AMPure PB Beads, size-selecting using the BluePippin System, and repairing DNA damage after size-selection. Library quality was assessed using the Qubit 2.0 Fluorometer (Thermo Scientific), and the insert fragment size was detected by the Agilent 2100. The preliminary assembly of the genome sequence was conducted with SMRT Link v5.0.1 and corrected using the arrow algorithm with the Variant Caller module of the SMRT Link software. The chromosomal sequence was confirmed to form a circle based on the overlap between the head and the tail. The initial site was then corrected by comparing it with the genome sequence of *Mr. extorquens* (CP001510.1). After processing the data, the sequences underwent reads mapping, SNP/Indel analysis using SAMTOOLS v.0.1.18, structural variation analysis, whole-genome variation mapping, and copy-number variation analysis using CNVnator v.0.3.

### RNA isolation and real-time reverse transcription-quantitative PCR (RT-qPCR) analysis

RNA isolation and RT-qPCR analysis followed previously described methods[29]. In brief, total RNA was extracted from 3 mL of cells in the mid-exponential phase using the TransZol Up RNAspin Mini Kit (TransGen Biotech, Beijing, China), and residual genomic DNA was removed using DNase I (Vazyme Biotech Co., Ltd., Nanjing, China). Purified RNA was then utilized for RT-qPCR, where 1000 ng RNA was reversely transcribed into cDNA using the HiScript II QRT SuperMix kit for qPCR (Vazyme Biotech Co., Ltd., Nanjing, China). Two-step real-time PCR was performed on an ABI 7500 system with the AceQ Universal SYBR qPCR Master Mix. The qPCR cycling conditions were as follows: 1. 95 °C for 5 min; 2. 95 °C for 10 s; 3. 60 °C for 34 s; 4. Go to 2 for 40 cycles; 5. 95 °C for 15 s; 6. Melt curve, 60 °C to 95 °C (increment 0.15 °C/s) for 1 s. The gene *rpsB*, encoding ribosomal protein S2, served as the reference gene. The average threshold cycle (Ct) value for each gene was calculated using a previously described method[29]. The $\Delta$Ct value indicated the relative transcriptional levels of target genes to the reference *rpsB* gene, while the $\Delta\Delta$Ct value described the difference between the transcriptional levels of different strains. The difference in expression was calculated using the $2^{-\Delta\Delta Ct}$ method.

### Expression, purification, and activity assay of enzymes

The genes *prs* (META1_4249) encoding PRS and META1_3141 were introduced into *E. coli* BL21 (DE3) via transformation using the plasmid pET32a to facilitate overexpression. The recombinant *E. coli* BL21 (DE3) was cultivated in LB broth medium at 37 °C until reaching an $OD_{600}$ of 0.6. Subsequently, induction was performed with 0.5 mM isopropyl β-D-1-thiogalactopyranoside (IPTG) for 14 h at 16 °C. Cell pellets were obtained through centrifugation, resuspended in lysis buffer, and lysed using the One-Shot Press (Constant Systems Cell Disruptor, Constant Systems, UK). The resulting recombinant proteins were harvested and purified via affinity chromatography using a Hi Trap affinity column following the manufacturer's guidelines (GE Healthcare, Piscataway, NJ, USA). The purity of the recombinant proteins was assessed using sodium dodecyl sulfate polyacrylamide gel electrophoresis (SDS-PAGE) and confirmed to exceed 90%. The enzymatic activity of PRS was determined using the previously described cascade method[67]. The assay involved monitoring the decline in NADH, which was indicative of lactate formation. The formation of 5-phosphoribosyl 1-pyrophosphate was assessed using myokinase, pyruvate kinase, and lactate dehydrogenase. The reaction mixture comprised 50 mM $KH_2PO_4$-$K_2HPO_4$ buffer (pH 7.9), 150 mM NaCl, 0.25 mM DTT, 2 mM phosphoenolpyruvate, 5 mM ATP, 1 mM NADH, two units of adenylate kinase, two units of pyruvate kinase, and two units of lactate dehydrogenase. The reaction was

initiated by adding 5 mM ribose 5-phosphate at 30 °C. After a 15-min incubation period, the decrease of NADH was determined as previously described. The enzyme activity was quantified in units, with one unit defined as the amount of enzyme catalyzing the formation of 1 μmol of PRPP per minute. Moreover, the three-dimensional structure of PRS was modeled using AlphaFold2 software[68].

The activity of FDH was assessed in cell crude extracts obtained through the One-Shot Press system, following the established protocol[69]. The reaction mixture consisted of 50 mM tricine-KOH (pH 7.0), 30 mM sodium formate, 0.5 mM $NAD^+$, and an appropriate amount of proteins. Enzymatic assays were conducted at least in triplicate. One unit of the enzyme activity was defined as the amount of enzyme oxidizing 1 μmol NADH or NADPH per minute.

### Extraction and analytical methods of metabolites

The *Mr. extorquens* strains were cultured on methanol at a concentration of 10 mM. 20 mL of cultures at the exponential phase ($OD_{600}$ of 0.4 ± 0.05) were rapidly harvested by vacuum filtration through 0.22-μm nylon membrane filters, and the collected strains were promptly quenched with liquid nitrogen, following established procedures[70]. Two samples (the AM1$^{PTR}$ and AM1$^{WT}$ strains) were analyzed, with three biological replicates analyzed for each sample. For the extraction of most intracellular metabolites, 10 mL of boiling water was employed, with subsequent incubation at 100 °C for 10 minutes, as detailed previously[71]. The resulting mixture was then centrifuged at 14,000 × *g* for 20 min at 4 °C, and the supernatant was collected and subjected to drying in a freeze drier (Christ ALPHA 1-2 LD plus, Germany). The dried sample was re-dissolved in 100 μL acetonitrile/water (1:1, v/v) solvent and centrifuged at 14,000 × *g* at 4 °C for 15 min. For the extraction of temperature-sensitive metabolites, 900 μL of cold methanol/acetonitrile/water (2:2:1, v/v) extraction solvent was added to the samples and thoroughly vortexed. The resulting suspension was sonicated in an ice bath for 20 min using a sonicator with a power of 150 W at 25 kHz (Scientz JY92-IIN, Ningbo, China). Stable-isotope internal standards were introduced into the extraction solvent concurrently to rectify variations arising from sample extraction and injection. LC-MS analyses were conducted using a UHPLC system (1290 Infinity LC, Agilent Technologies) coupled with a QTRAP MS (6500+, AB SCIEX). Metabolites were separated on both a HILIC column (Waters UPLC BEH Amide column, 2.1 mm × 100 mm, 1.7 μm) and an RPLC column (Waters UPLC BEH C18 column, 2.1 mm × 100 mm, 1.7 μm). For HILIC separation, the column temperature was set at 35 °C, and the injection volume was 2 μL. The mobile phase comprised 90% $H_2O$ + 2 mM ammonium formate +10% acetonitrile for phase A and 0.4% formic acid in methanol for phase B. A gradient program was employed (85% B at 0-1 min, 80% B at 3-4 min, 70% B at 6 min, 50% B at 10−15.5 min, and 85% B at 15.6−23 min) at a flow rate of 300 μL/min. For RPLC separation, the column temperature was maintained at 40 °C, and the injection volume was 2 μL. The mobile phase included 5 mM ammonium acetate and 0.2% $NH_3\cdot H_2O$ in water for phase A and 99.5% acetonitrile +0.5% $NH_3\cdot H_2O$ for phase B. A gradient program was initiated (5% B at 0 min, 60% B at 5 min, 100% B at 11-13 min, 5% B at 13.1−16 min) at a flow rate of 400 μL/min. QTRAP MS was operated in positive and negative switch mode. The ESI positive source conditions were as follows: source temperature of 580 °C; ion source gas 1 of 45; ion source gas 2 of 60; curtain gas of 35; ion spray voltage of +4500 V. The ESI negative source conditions were as follows: source temperature of 580 °C; ion source gas 1 of 45; ion source gas 2 of 60; curtain gas of 35; ion spray voltage of −4500 V. Multiple reaction monitoring (MRM) was employed for mass spectrometry quantitative data acquisition. The MRM ion pairs are presented in Supplementary Data 1. All peaks were analyzed using MultiQuant software (v3.0.3) for quantitative data processing. The acid derivatives of mesaconyl-CoA, methyl-succinyl-CoA, and 3-hydroxybutyryl-CoA were subjected to analysis using GC-MS (GC17A with QP-2020 mass spectrometer, Shimadzu,

Kyoto, Japan), following established procedures[29]. In brief, the sample underwent drying and methoxylation by adding a methoxyamine solution, followed by incubation at 60 °C for 30 min. Subsequently, 50 µL of TMS reagent was introduced for trimethylsilylation, and the mixture was further incubated at 30 °C for an additional 30 minutes. The derivatized sample was separated on an Rtx-5MS column (30 m × 0.25 mm × 0.25 µm, Restek, USA). The inlet and transfer line temperatures of GC-MS were set at 280 °C. The temperature program was initiated at 60 °C with a hold time of 0.25 min, followed by an increase to 280 °C at a rate of 5 °C/min with a hold time of 10 min at 280 °C. The ion source and quadrupole temperatures of GC-MS were set at 250 and 150 °C, respectively. Mass spectra were collected from m/z 40–500 after a solvent delay of 4.5 min. All peaks were analyzed using GCMS LabSolutions software (v4.45 SP1).

PHB extraction and analysis were performed following previously established protocols with slight modifications[72]. The instrumental analysis employed a GC-MS system (GC17A coupled with a QP-2020 mass spectrometer, Shimadzu, Kyoto, Japan) equipped with an SH-Rxi-5Sil MS capillary column (30 m × 0.25 mm × 0.25 µm, Shimadzu, Japan). The temperature program commenced at 50 °C for 5 minutes, followed by an increase to 300 °C at a rate of 30 °C/min. Extracellular IAA production was quantified using Salkowski's reagent[73]. ACC deaminase activity was assessed by measuring the formation of 2-ketobutyrate[39]. Equal volumes of the reaction mixtures containing ACC deaminase produced by the same amount of RA$^{PTR}$ and RA$^{WT}$ cells were employed. The total ACC deaminase activity from the RA strains was reflected by the quantified amount of 2-ketobutyrate.

### Colonization assay of Methylobacterium/Methylorubrum on A. thaliana leaves

*Methylobacterium/Methylorubrum* strains were cultured in liquid MSM containing 120 mM methanol until the early stationary growth phase. To evaluate their colonization ability on *A. thaliana* leaves within a gnotobiotic system, the strains were harvested, washed once, and resuspended in a sterile 10 mM MgCl$_2$ solution. The cultures were then adjusted to an OD$_{600}$ of 0.5, and 5-µL aliquotes were pipetted onto each seed during sowing. For evaluating colonization efficiency under two-strain competitive conditions, 2.5 µL of the PRS mutant strain or its WT counterpart strain suspension (OD$_{600}$ = 0.5) was pipetted onto each seed along with their respective strain. *A. thaliana* Col-0 plants were cultivated on standard Murashige and Skoog (MS) nutrient agar medium supplemented with 3% sucrose. The autoclaved medium was poured into sterile tissue-culture containers, and *A. thaliana* seeds were surface-sterilized and placed on the agar surface[20]. Under long-day conditions of 16 h of light and 8 h of darkness, plants were grown in sterile tissue-culture containers in a standard growth chamber at a constant temperature of 22 °C. Axenic plants were mock-treated with a sterile 10 mM MgCl$_2$ solution. After 21 days of growth, the population of *Methylobacterium/Methylorubrum* on *Arabidopsis* leaves was determined by randomly selecting plants from a group of over 25 replicates[74].

To evaluate the colonization ability of the PRS mutant strain of *Mr. extorquens* on *A. thaliana* leaves within growth chambers, *A. thaliana* plants were allowed to grow for 7 days prior to foliar spraying with 1 mL of culture liquid per plant. The culture liquid contained either the PRS mutant strain or its WT counterpart. Specifically, the culture liquid consisted of 1 mL of a 10 mM MgCl$_2$ solution containing 10$^6$ CFUs of *Mr. extorquens* cells. Colonized populations were assessed on days 4, 8, and 14 following foliar spraying, using established methods[75]. To avoid damaging plant growth and to measure the colonized cell numbers as accurately as possible, each experimental group was divided into four subgroups, and in each subgroup, five leaves were randomly selected from different plant individuals and mixed to determine the number of colonized cells, resulting in four independent biological subgroups.

Total DNA was then extracted from each independent biological subgroup using the MagMAX Plant DNA Isolation kit, following the manufacturer's instructions (Thermo Fisher Scientific). The unique genomic regions of *Mr. extorquens* were identified from the sequence in GeneBank (CP001510.1, 1631527..1632060), and corresponding primer pairs were employed to ensure adequate specificity. qPCR was performed using an ABI 7500 system with the AceQ Universal SYBR qPCR Master Mix. The PCR cycling conditions were as follows: 1. 95 °C for 5 min; 2. 95 °C for 10 s; 3. 60 °C for 34 s; 4. go to 2 for 40 cycles; 5. 95 °C for 15 s; 6. Melt curve, 60 °C to 95 °C (increment 0.15 °C/s) for 1 sec. The number of *Mr. extorquens* cells associated with *A. thaliana* leaves was quantified using a standard curve constructed from cell suspensions of *Mr. extorquens*. The Ct values from the qPCR data for each serial dilution were plotted against the log value of the number of *Mr. extorquens* cells in the serial dilution to obtain the standard curve.

### Assay for promoting plant growth and yield through foliar spraying with *Methylobacterium* strains

To investigate the impact of the PRS mutant strain of *Mb. radiotolerans* JCM2831 on the growth promotion of *C. melo, C. sativus*, and *B. chinensis* seedlings, seedlings were cultivated in growth chambers until they reached 12 days of age. Subsequently, they were sprayed with either the PRS mutant strain or its WT counterpart at a concentration of 4 × 10$^4$ CFUs per milliliter. Each plant was sprayed with a 1-mL culture. Colonized populations were quantified using qPCR targeting a unique genomic region (GeneBank: CP001001.1, 1221888..1223114) on days 7 and 12 post foliar spraying, as described before. Additionally, the fresh weight of each plant seedling was measured on day 12 post foliar spraying, providing an assessment of the growth-promoting effects of the PRS mutant strain on *C. melo, C. sativus*, and *B. chinensis* seedlings.

The impact on *B. chinensis* by spraying either the RA$^{PTR}$ or the RA$^{WT}$ strain was evaluated in a standard Chinese agricultural solar greenhouse located in Jimo District, Qingdao City, Shandong Province, China (North latitude 36°22' and East longitude 120°22'). The greenhouse maintained a temperature of approximately 20 °C during the day and 10 °C at night, with humidity levels ranging from 50% to 60%. The soil conducive to plant growth had a composition detailed in Supplementary Table 4. The experiment was conducted from December 24, 2023, to February 03, 2024. *B. chinensis* seeds were planted in a solar greenhouse with a row spacing of 30 cm and a plant spacing of 25 cm. After growing for 17 days and 28 days, each plant received a 1-mL culture of either the RA$^{PTR}$ or RA$^{WT}$ strain (4 × 10$^4$ CFU/mL) via foliar spray. The control group was sprayed with 1 mL of MgCl$_2$ solution (10 mM) (Supplementary Fig. 14). The colonized population was quantified using qPCR on days 4, 8, and 11 post the latter foliar spray, following the previously described method. Once *B. chinensis* had reached the appropriate size for the market readiness (i.e. 40 days old), it was harvested to determine the yield of each experimental group.

### Phylogenetic analysis

Genome sequences available on NCBI were analyzed to identify *Methylobacterium/Methylorubrum* species possessing the EMC and PKT pathways. Marker genes for the EMC pathway included *croR, ccr, ecm, msd*, and *mcd*, while *xfp, pta*, and *ack* served as markers for the PKT pathway. The presence of these marker genes in the genomes was used to infer the utilization of the EMC and PKT pathways. A phylogenetic tree was constructed based on 16S rRNA sequences using the neighbor-joining method in MEGA X. The phylogenetic tree was visualized using iTOL (https://itol.embl.de/)[76]. This approach allowed for the identification and classification of *Methylobacterium/Methylorubrum* species based on their evolutionary relationships and the presence of specific metabolic pathways.

## Statistical analysis

Statistical analyses were mostly performed using Microsoft Excel software (version 2021), employing a two-tailed t-test or one-way ANOVA with posthoc tests for analysis of variance. The data were presented as mean ± standard deviation (SD). Each panel comprised a minimum of at least three independent biological samples. Significance was established with a $p$-value less than 0.05 compared to a reference sample.

## Reporting summary

Further information on research design is available in the Nature Portfolio Reporting Summary linked to this article.

## Data availability

Whole-genome sequencing data of the strains of RSE01, RSE02, and RSE07 have been deposited at the National Center for Biotechnology Information under Bioproject number PRJNA1031812. Source data are provided with this paper.

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

## Acknowledgements

This study was financially supported by National Key Research and Development Program of China (2018YFA0901500 to X.H.X. and 2021YFC2103500 to S.Y.), National Natural Science Foundation of China (22078169 to S.Y. and 31900004 to Cong Z.), Shandong Provincial Key Research and Development Plan (2021ZDSYS28 to S.Y.), and the U.S. Department of Energy under the DOE Office of Science (SC) RENEW contract (DE-SC0024289 to M.G.K.).

## Author contributions

S.Y. and Cong Z. designed the experiments. D.F.Z., M.Y.W., Y.Z.S., M.M.Z., L.Y., X.J.Y. and J.S. performed the experiments. S.Y., Cong Z., D.F.Z., Y.Z.S., L.Y., X.H.M., Z.X.M., Y.S., H.R.W., S.H.D. and S.H.L. analyzed the results. S.Y., Cong Z., Chong Z., L.Y., K.B., M.S., M.G.K. and X.H.X. wrote the manuscript. All authors edited the manuscript before submission.

## Competing interests

The authors declare no competing interests.

## Additional information

[1]School of Life Sciences, Qingdao Agricultural University, Qingdao, Shandong, PR China. [2]Shandong Province Key Laboratory of Applied Mycology, Qingdao Agricultural University, Qingdao, Shandong, PR China. [3]Qingdao International Center on Microbes Utilizing Biogas, Qingdao Agricultural University, Qingdao, Shandong, PR China. [4]Key Laboratory of Industrial Biocatalysis, Ministry of Education, Department of Chemical Engineering, Tsinghua University, Beijing, PR China. [5]Center for Synthetic and Systems Biology, Tsinghua University, Beijing, PR China. [6]Tobacco Research Institute, Chinese Academy of Agricultural Sciences, Qingdao, Shandong, PR China. [7]School of Life Sciences, Hubei University, Wuhan, Hubei, PR China. [8]CABIO Biotech (Wuhan) Co. Ltd., Wuhan, Hubei, PR China. [9]Department of Chemistry, University of Washington, Seattle, WA, USA. [10]Department of Biology, San Diego State University, San Diego, CA, USA. [11]Institute of Biopharmaceutical and Health Engineering, Tsinghua Shenzhen International Graduate School, Shenzhen, PR China. [12]Institute of Biomedical Health Technology and Engineering, Shenzhen Bay Laboratory, Shenzhen, PR China. [13]School of Marine Science and Engineering, Qingdao Agricultural University, Qingdao, Shandong, PR China. [14]Key Laboratory of Systems Bioengineering, Ministry of Education, Tianjin University, Tianjin, PR China. [15]These authors contributed equally: Cong Zhang, Di-Fei Zhou, Meng-Ying Wang, Ya-Zhen Song. ✉e-mail: yangsong1209@163.com; SongYang@qau.edu.cn

