## [Peer Review File · Nature Communications]

REVIEWER COMMENTS

Reviewer #1 (Remarks to the Author):

The authors derived strains with higher growth activity during methylotrophic growth through adaptive evolution from several *Methylobacterium* strains (with or without heterologous RuMP pathway /the original serine pathway). After the genome sequences of several *Methylobacterium* strains, transcriptome analyses, the authors identified phosphoribosylpyrophosphate synthetase (PRS) is a key factor for methylotrophic growth. And RPS-modified strains were tested for colonization efficiency on *A. thaliana*, and effect on seedling growth following foliar spray with a low dose, and they demonstrated that PRS mutation could give positive effect on both colonization efficiency and plant growth promotion activity.

The manuscript contains some interesting observation. In spite of the authors' efforts, improvement of the methylotrophic growth in PA1PTR and PA1WT is very small (Fig. 4C). And the authors' model is not confirmed by biochemical analyses: how does the metabolism changed the growth of *Methylobacterium*? Why does PTR mutation resulted in improvement of colonization efficiency and growth seedling growth of the host plant? Since other mutations could occur during adaptive revolution, improvement of methylotrophic growth cannot be attributed to the only reason for these positive effects on plant.

From biotechnological aspects, the observed phenomena observed with *Arabidopsis* cannot be applied simply to other crops, and for the moment, genetically engineered microbial strains could not be used for agriculture. Since growth improvement is very small, the use of RPS mutant strains do not have merits for production of useful chemicals.

In summary, this manuscript lacks impacts sufficient for publication in *Nat Communications*.

Another comment:

The first part with RuMP pathway-introduced strains ends with negative results, and this section is very confusing for readers. The authors should remove on the topic or refer to only briefly.

Reviewer #2 (Remarks to the Author):

Zhang and colleagues report here about a change in the metabolism of *Methylobacterium* and *Methylorubrum* strains that leads to improved growth at low methanol concentrations, which turns

out to be advantageous for phyllosphere colonization. Their findings result from comprehensive research, beginning with a laboratory evolution experiment along with targeted genetic modifications, resulting in a strain with improved growth under low methanol concentrations. This is followed by genomic resequencing and mutation/cultivation experiments to identify and validate the genetic basis and the underlying metabolic changes for the improved performance. Next, the effect was proven to exist in further strains of the *Methylobacterium*/*Methylorubrum* group when introducing the same type of mutation. In the last part of the work, the authors show a positive effect on phyllosphere colonization under climate chamber conditions, which is accompanied by increased plant seedling growth. This leads the authors to the conclusion that the identified mechanism can lead to improved plant colonization and represents a lever for more effective phyllosphere colonization of this microbial group, which can go along with improved plant growth.

I consider this an extensive and solid work with very interesting findings. However, I have one major criticism, which is related to the discussed implications the findings may have for crop management. As many studies, this work ends by showing positive effects on plant growth with plants grown in a climate chamber. However, as also stated by the authors, the major challenge when studying plant growth promoting effects of microorganisms is the transfer of lab-based observations to the field. This important validation step is not included in the present work. Even though the results regarding the role of the altered PRS enzyme for efficient plant colonization are convincing, the usefulness for improving the colonization efficiency of *Methylobacterium*/*Methylorubrum* strains as biologicals under field conditions remains largely open. Under field conditions, we can expect a stronger competition between *Methylobacterium*/*Methylorubrum* strains with other phyllosphere colonizers. Climate chamber plants usually host a substantially lower abundance of bacteria compared to plants grown under field conditions. Consequently, other resources than carbon may be the limiting factor for methylootrophs under field conditions and define the population size of the methylootrophs under these conditions. That the altered traits will indeed lead to an increased population size under such conditions demands further validation. This is further substantiated by the results shown in Fig. 6d, where the differences between wildtype and mutant appear to decrease with increasing plant age. Therewith, the relevance for the improvement of the plant growth potential remains also rather unclear at this stage. The authors should either discuss this much more clearly and be more careful regarding the implications for crop management, or provide further evidence.

Besides this point, other issues should be addressed prior to publication.

The applied statistical tests are not always adequate. Even if the results are convincing and will most likely not change, the statistical analysis should be done correctly. The authors report that they performed t-tests in Excel. Considering that they often analyzed CFU numbers, data are not necessarily normally distributed and may request log-transformation. Was this evaluated? Further, the data in Fig. 5d and 6d request ANOVA with posthoc tests, not multiple t-tests.

It is said that all results are reported based on at least three replicates, often indicated as “n larger or equal to three”. This is fairly unspecific and can in principle be any number larger than three. Please specify the min and max number of replicates of a dataset if not uniform. Besides, the information about replicates given in figure legends appears to be inappropriate in some cases.

The authors created complex figures with several diagrams, which are very small in size. It is very hard and in part impossible to read the text in these figures on a print out or without substantial page enlargement, especially when diagrams are shown as inserts. Even though articles are nowadays mostly read as pdfs, where figures can be enlarged, a certain size should be provided to enable a reader to quickly grasp and understand the figures. Authors should follow the guidelines.

The authors should refer correctly to the genera *Methylobacterium* and *Methylorubrum*. Upon reclassification some years ago, these are two different genera and they studied members of both.

Further, the species names of *M. radiotolerans* and *M. oryzae* are frequently misspelled. This needs a systematic correction including their mentioning in text, figures, images and tables (incl. supplementary material).

Besides, the names of the strains should be given that were used in this work. This applies to *M. radiotolerans* and *M. oryzae*.

The authors state that some findings and conclusions apply to methylotrophic Alphaproteobacteria, but this claim cannot be made, because they only studied a specific clade, i.e. *Methylobacterium*/*Methylorubrum*. Besides these, other methylotrophic Alphaproteobacteria exist, which were not part of this study.

The manuscript should be proof-read by a native speaker. There are a couple of unclear statements and different grammatical issues. This is somewhat surprising considering that two co-authors are affiliated to Universities in USA and are perhaps native speakers. If not, the manuscript should undergo professional proof-reading. I highlight some issues below to demonstrate the problem, but do not consistently point out all issues.

Some abbreviations are introduced twice in different sections (e.g. ALE). In figure legends, it would be more intuitive to give abbreviations/names in brackets rather than the corresponding explanations (applies to strain derivatives).

Specific comments:

Abstract:

l. 32: rephrase; *Methylobacterium* is not a microbiome, but part of the phyllosphere microbiome

l. 34: it is not proven that *Methylobacterium* is the predominant symbiotic partner of plants in the phyllosphere; it has been reported to be a dominant colonizer of the phyllosphere and strains have been demonstrated to be able to promote plant growth, but whether this genus is indeed the “predominant symbiotic partner that promotes plant growth within the phyllosphere” is not known in general and under field conditions. Rephrase and be more precise regarding this statement.

l. 39-40: You state here that you worked with *M. extorquens* AM1 strains being capable of methanol utilization via a heterologous ribulose monophosphate pathway. Can you be sure that this pathway was indeed used? You observed that the plasmid carrying RuMP genes could be removed (after experimental evolution) without problems. This questions whether the pathway was needed at all. Besides, this information is not given in the results section.

l. 41: You write here that the serine cycle was restored, but it is not clear to the reader that it was missing or defect at this point of the abstract.

l. 43: add “the” and “gene”, i.e. “in the phosphoribosylpyrophosphate synthetase (PRS) gene was”

l. 50: “offer”, not “offered”

Introduction:

l. 84: the species *M. extroquens* is not necessarily a representative of the *A. thaliana* leaf microbiota, this only applies to specific strains of this species. Add strain name to specify your statement.

l. 96: you probably mean plant growth promoting here, not plant promoting.

Results:

l. 111: “more efficient” requests a comparative statement; part of the comparison is missing here; it needs to be stated to which alternative pathway the efficiency of the RuMP pathway is compared.

l. 115: please add a short sentence to say which genes of the RuMP genes were introduced and how.

l. 128-130: two points to this statement: 1.) in case all SE lineages activated the mentioned hypothetical protein, I would point this out more clearly, because the presented results show different performance of different strains; i.e.: add “all” 2.) it would be useful to add a short sentence explaining how this was proven (based on which approach).

l. 131: add the, i.e. “in the plant phyllosphere”

l. 135: please specify that this was done via overexpression on a plasmid.

l. 137 and elsewhere (especially figure legends): what does “typical methanol” mean? For bacteria under natural conditions 120 mM may not be typical at all; I recommend referring to “standard cultivation conditions” or something alike.

l. 145: here and elsewhere, please state correctly “on low methanol concentrations”; i.e. add the word concentration

l. 156 correct by writing “to an asparagine residue”

l. 177-180: The observation reported in this statement cannot be seen in Fig. 2e. First, I see a clear difference also in the presence of 1.2 mM methanol, but at the same time, data comparison is difficult, because the growth performance under higher methanol concentrations is illustrated based on differences in OD, while the performance under the low concentrations is illustrated based on CFU counts. I don’t understand why different types of data are shown for different concentrations, this should be uniform to allow direct comparison of responses to all conditions. This may explain my disagreement with your statement?

l. 245: It would facilitate reading if the authors would briefly point out the role of *xfp2* here again.

l. 251: Reference should be to Fig 3e here, not 3f.

l. 259: It would be valuable to provide information about the identity of all 64 strains to know which ones were different, e.g. in supplement.

l. 289: Do the authors refer to leaf fresh weight or plant fresh weight here?

l. 300: Do the authors know for sure that the mutation does not impair the growth of *M. extorquens* in the phyllosphere? Has this been experimentally proven? If so, provide reference.

l. 307-308: You may want to specify the purpose of this experiment here.

l. 308-310: The sentence is confusing and probably not correct, because in Fig. 5e results from foliar application are shown, while it is said here that seeds were treated.

l. 320-324: phrasing unclear.

l. 333: “visible abundance” does not appear to be a meaningful phrase; what would that be?

l. 335: “open climatic chamber”? What do the authors mean with this?

l. 338: specify what the control group was.

Discussion:

l. 416: add reference to support this statement

l. 429-431: not clear how this can be realized, because to my understanding it needs genetic engineering of the *Methylobacterium* strains to have the observed improved performance.

Methods:

The genomic sequencing part is not explained with sufficient detail to reproduce.

Check the correct formatting of gene names (e.g. l. 557, 561 and possibly elsewhere).

qPCR cycling conditions should be described to enable others to reproduce it.

l. 595-598: sentence grammatically incorrect; rephrase

l. 652: provide protocol for *A. thaliana* seed sterilization or give an appropriate reference

l. 661 and 673: how to know that this number of CFUs was applied per plant? Is it the theoretical maximum number? Not all cells will reach the leaf surface when spraying I assume. Better to provide in addition information on the cell density of the suspension that was applied and the amount of liquid that way sprayed per plant.

Figures/tables:

l. 897: correct formatting of WT

Fig. 3e: Is “NA” correct (not analyzed, not applicable, not available?); shouldn’t it be “n.s.” (not significant).

Fig. 4b: It could have been interesting to evaluate whether the 34 strains represent specific subgroups within the *Methylobacterium*/*Methylorubrum* diversity, but as only the 34 strains are

shown, I don't see the relevance of this tree. But this appears not to be the case, so an extended tree will probably not add anything, so I recommend removing it. Further, the legend does not indicate based on which gene(s) the tree was calculated and based on which algorithm, not does it explain the red dots behind some strains. More important would be the identity of all strains included in the genomic analysis in the supplement.

Fig. 5 a-c: It is unclear what replicate 1 and replicate 2 means in this context, assuming that each dot is representing the CFU determined on one plant individual, which would in my understanding represent a replicate. Besides, the statistics with such low p-values were probably not done based on two replicates only. Either the definition of replicates is inappropriate, the statistics, or both.

Fig. 5e: this figure reports qPCR results, the y-axis labelling with CFU is incorrect. It should report gene copy numbers, if a specific gene was targeted. Otherwise, adjust accordingly.

Fig. 6 b and c: at which day / plant age were these images taken and the data determined?

Fig. 6: legend unclear by stating that data are based on n = 3 biologically replicates, I see already 6 plant individuals in the photographs and many dots in panel c.

Files with supplementary figures and tables should be declared as such, in the file name (if possible) and within the file. For the tables given in excel files, it would be useful to add the table header within the respective data file rather than in a separate document.

Fig S2: I don't see a need to show Fig. 2a, because this did not result in any strains that were further studied here.

Fig. S3 would profit from a bit more detailed legend.

Table S6: please provide information about target genes and/or target organism if specific.

Reviewer #3 (Remarks to the Author):

Zhang and collaborators demonstrate that the partial loss of the phosphoribosylpyrophosphate synthase (PRS) function discretely but significantly improved *Methylobacterium* growth in conditions of limited carbon sources, and this loss enhanced plant colonization. This finding is original and of great interest to both fundamental and applied scientific communities.

Comments:

1. Although methanol was the substrate used for the discovery that PRS is a metabolic valve that enhances *Methylobacterium* phyllosphere colonization, this process might not be restricted to methanol as shown in Figure 1. The effect of the PRS mutation on *Methylobacterium* growth, a facultative methylotroph, was observed for methanol and other multi-carbon sources, each studied alone as the only carbon source, but no combination of limiting amounts of different

carbon sources was assessed together in this study (although low concentration of methanol and multi-carbon compounds are found in the phylloplane).

2. A high methanol concentration of 120mM is not a 'typical' concentration used in labs working with *Methylobacterium* strains. The point here is that this concentration provides methanol in excess and is not growth-limiting. Revise the text in consequence throughout the manuscript (L117, L124, etc.).

3. Few *Methylobacterium* species have been described as strong symbionts, which may not be the case of the species studied in this work (except for *M. nodulans*). To avoid misleading the reader, consider modifying the text in the abstract (L34).

4. Do the strains used in this study have the Δcel mutation that prevents cell aggregation in lab cultures? If not, this may affect DO and CFU proportionality measurements with biomass. Did the cells aggregate in this study?

5. For the study of the PRS mutation effect on *A. thaliana* leaf colonisation from seed inoculation, the experiments were carried out with *M. extorquens* AM1. Why was *M. extorquens* AM1 chosen, rather than PA1, which is known to be a better colonizer? In Fig. 5e, provide the qPCR data for the detection of *M. extorquens* AM1 population on *Arabidopsis* leaves, in addition to viable count cells. How can CFU be safely assigned to *M. extorquens* AM1? How were the AM1 specific primers validated?

6. After spraying *M. radiotolerans* WT strain or PRS mutant on different plant seedlings, how do you assign CFU to *M. radiotolerans* and distinguish them from native epiphytes (Fig 6d; L342-344)?

Other comments:

Fig. 3c. For clarity, specify in the legend that version 04 indicates a mutated *xfp2* gene.

Fig. 3e. Why classify "His" under nucleotides?

Fig. 4a. Specify the "other *Methylobacterium* species"

Fig. 4b. Specify the identity of all of the *Methylobacterium* strains used in the phylogenetic analysis.

Fig. 6b. No size ruler was shown in the different pictures.

Supplemental Fig. 1. Indicate color code

Supplemental Fig. 3. Correct "*M. radiotolarance*". What is the color code? Correct the text covered by arrows.

AUTHORS RESPONSES TO THE REVIEWERS COMMENTS.

Reviewer #1 (Remarks to the Author):

The authors derived strains with higher growth activity during methylotrophic growth through adaptive evolution from several *Methylobacterium* strains (with or without heterologous RuMP pathway /the original serine pathway). After the genome sequences of several *Methylobacterium* strains, transcriptome analyses, the authors identified phosphoribosylpyrophosphate synthetase (PRS) is a key factor for methylotrophic growth. And RPS-modified strains were tested for colonization efficiency on *A. thaliana*, and effect on seedling growth following foliar spray with a low dose, and they demonstrated that PRS mutation could give positive effect on both colonization efficiency and plant growth promotion activity. The manuscript contains some interesting observation.

We greatly appreciate your positive feedback!

In spite of the authors' efforts, improvement of the methylotrophic growth in PA1PTR and PA1WT is very small (Fig. 4C).

Response: As shown in Fig. 4C (below is an enlargement of growth curves of *Methylobacterium extorquens* PA1 in Fig 4C), the PRS mutation reduced the lag phase and increased the maximum OD₆₀₀ in *Methylobacterium extorquens* PA1. The lag phase of the PA1^{WT} strain was approximately 4 hours grown at low methanol concentrations of 5 mM, 10 mM and 15 mM, while that of the PA1^{PTR} strain was 3 hours or less. The maximum OD₆₀₀ of the PA1^{PTR} strain increased by over 10 % at 5 mM and 10 mM methanol, and approximately 7 % at 15 mM methanol. Additionally, the colonized populations of the PA1^{PTR} strain on *Arabidopsis* leaves under a gnotobiotic plant growth system were increased by 59% compared to that of the PA1^{WT} strain (Fig. 5b), indicating that the PRS mutation enhanced *Mr. extorquens* PA1 colonization in *Arabidopsis* phyllosphere effectively.

And the authors' model is not confirmed by biochemical analyses: how does the metabolism changed the growth of *Methylobacterium*?

Response: Thank you very much for the comment. In our research, a single nucleotide mutation in the phosphoribosylpyrophosphate synthetase (PRS) gene was identified, which converted it into a metabolic valve. This valve led to trade-offs between nucleotide anabolism and central assimilatory metabolism through the up-regulated phosphoketolase (PKT) pathway in *Methylobacterium/Methylorubrum* strains. This metabolic reprogramming redirected limited C1-carbon resources towards biomass synthesis, resulting in more competitive growth at low methanol concentrations. The biochemical evidence is as follows:

(i) Metabolites analysis in the *Mr. extorquens* strains AM1^{PTR} and AM1^{WT} grown at low methanol of 10 mM indicated that partial impairment of PRS function resulted in many nucleotides decrease in pool size in the AM1^{PTR} strain (Fig 3a). Histidine and NADP pools showed a similar pattern to the nucleotides. Furthermore, acetyl-phosphate, which is readily converted to acetyl-CoA through the PKT pathway, was significantly higher in the AM1^{PTR} strain, and accordingly, the substrate fructose-6-phosphate required for the PKT pathway was greatly reduced. These metabolites analysis suggested that partially reduced PRS activity limited the diversion of F6P into PRPP biosynthesis, while concurrently promoted the PKT pathway, contributing to increased growth fitness at low methanol concentrations.

(ii) Transcripts analysis showed that the genes *xfp* (*xfp1*, *xfp2*), *pta* and *ack* (*ack1*, *ack3*) involved in the PKT pathway were significantly up-regulated in the AM1^{PTR} strain compared to the AM1^{WT} strain grown on 10 mM methanol (Fig 3b). Genes associated with the EMC pathway, including *phaA*, *phaB*, *phaC*, *mcmAB*, and *mcl*, as well as genes involved in the H₄F-dependent formate transfer pathway and the serine cycle (*ftfl*, *fch*, *ppc* and *mdh*) and genes in the gluconeogenic pathway (*gapA*, *pgk*, and *cbbA*) also exhibited increase in expression levels when comparing the AM1^{PTR} strain with the AM1^{WT} strain. Furthermore, the transcription of *fdh1234* encoding formate dehydrogenase (FDH) was down-regulated by 0.3- to 0.7-fold in the AM1^{PTR} strain. Accordingly, the FDH activity decreased by 17% in the AM1^{PTR} strain (Fig 3c). These declines appeared to coincide with a formate branchpoint redistribution that complemented the enhanced methanol assimilation.

(iii) The *xfp* gene plays a crucial role in the PKT pathway by catalyzing fructose-6-phosphate or xylulose-5-phosphate into erythrose-4-phosphate or glyceraldehyde-3-

phosphate and acetyl-phosphate. Knockout of *xfp2* or the interference with its transcription obviously reduced the growth advantages of the AM1^{PTR} strain on low methanol concentrations of 5 mM, 10 mM and 15 mM (Fig. 3d), but did not change the growth trend of the AM1^{WT} strain.

In summary, the above results indicate that partial impairment of PRS function led to a more prudent allocation of limited methanol resources, redirecting them away from nucleotide synthesis (Fig 3e). This adaptation effectively mitigates the direct competition between nucleotide anabolism and the core methylotrophic assimilation pathways, both of which are reliant upon H₄F derivatives and glycine, and reduces the energy requirements of nucleotide anabolism. Subsequently, the EMC pathway assimilated acetyl-CoA produced through the PKT pathway, facilitating increased CO₂ fixation and glyoxylate production. Glyoxylate was then condensed with 5,10-methylene-H₄F to expedite methanol assimilation via the serine cycle. Concomitantly, the up-regulation of central assimilation was accompanied by the down-regulation of complete formate oxidation. Consequently, this metabolic reprogramming provided a cost-effective and logical exchange, efficiently utilizing limited carbon, energy, and reducing equivalents for both nucleotide anabolism and the circulation of central metabolism. This led to an increase in biomass yields and conferred enhanced competitiveness in terms of growth and the colonization abilities of plants.

This comment made us realized that the description of the results is not complete. We modified our manuscript and added additional information on Pages 8-10.

Why does PTR mutation resulted in improvement of colonization efficiency and growth seedling growth of the host plant?

Response: The PRS mutation provided a growth advantage by shortening the lag phase and increasing the biomass in the AM1^{PTR} strain at low methanol concentrations (15 mM, 10 mM, 5 mM, 1.2 mM and 0.5 mM) as well as at low concentrations (1 mM and 0.1 mM) of multi-carbon sources such as acetate, ethanol, oxalate, pyruvate and succinate (Fig. 2e and i). As recommended by reviewer 3, we have investigated the growth of the AM1^{PTR} and AM1^{WT} strains on a combination of limited methanol (0.5 mM) and limited multi-carbon sources (containing 0.1 mM ethanol, 0.1 mM acetate, 0.1 mM oxalate, 0.1 mM pyruvate, and 0.1 mM succinate), mimicking the phyllosphere environment. Similar to the sole limited carbon source, this result indicated that the AM1^{PTR} strain also exhibited a growth advantage over the AM1^{WT} strain on combined carbon sources (Fig. 2j). The plant leaves release trace amounts of methanol and C₂ to C₄ substrates (Trends in Plant Science,1996,1: 296-301; Nat Rev Microbiol. 2012,10:828-840). The ability of phyllosphere microbes to effectively utilize plant-provided carbon nutrients is pivotal for their colonization of plants. On the basis of these, the PRS mutation in *Methylobacterium/Methylorubrum* strains is thought to improve their capacity to colonize the plant phyllosphere. This was then supported by the colonization results (Fig. 5a to e). Furthermore, the improvement in colonization efficiency was eliminated by knocking out the *xfp* gene in the AM1^{PTR} strain (Fig. 5f), suggesting that the PRS mutation improved the colonization efficiency through the activation of the PKT pathway in the plant phyllosphere, as in the AM1^{PTR} strain grown

under cultivation conditions.

The *Methylobacterium/Methylorubrum* strains have the capacity to improve plant growth and yield through the synthesis of auxin and 1-aminocyclopropane-1-carboxylate deaminase (ACC deaminase). We have investigated the production of indoleacetic acid (IAA) and ACC deaminase in the RA^{PTR} and RA^{WT} strains grown on methanol of 5 mM. As demonstrated in Supplementary Fig. 13 and below, the released amounts of IAA and ACC deaminase did not have any significant difference between the RA^{PTR} and RA^{WT} strains. This indicated that the PRS mutation did not affect the synthetic ability of IAA and ACC deaminase. It was likely that the higher plant growth-promoting ability of the PRS mutant strain was attributed to the higher number of colonized strains in the phyllosphere. The results were added on Page 14, lines 378-382.

Since other mutations could occur during adaptive evolution, improvement of methylotrophic growth cannot be attributed to the only reason for these positive effects on plant.

Response: We agree with the comment. It is possible that other mutations occurred during adaptive evolution may have played a role in the positive effects of evolutionary strains on plants. In our research, we introduced the *prs*^{EVO} allele (a single mutation of PRS) into *prs*^{WT} in the wild type *Mr. extorquens* AM1 to obtain mutant strain (named AM1^{PTR}), and then evaluated the effects of the wild type strain with the PRS mutation on growth at low methanol concentrations and on the plant colonization. This ensures that the positive effects on plants are solely due to the PRS mutation. We highlighted this in our manuscript on Page 7, Lines 177-184.

From biotechnological aspects, the observed phenomena observed with Arabidopsis cannot be applied simply to other crops, and for the moment, genetically engineered

microbial strains could not be used for agriculture. Since growth improvement is very small, the use of RPS mutant strains do not have merits for production of useful chemicals.

Response: Thank you for the suggestion! We agree with the comment. *Arabidopsis* is a model plant belonging to the Cruciferae family, and it can't simply be assumed that just because the PRS mutation works on *Arabidopsis* it can be applied to other crops. In our research, we have conducted colonization and seedlings growth studies on three other plants from two different families (*Brassica chinensis* of Brassicaceae family, and *Cucumis melo* and *Cucumis sativus* of Cucurbitaceae family) in a growth chamber (Fig. 6). These three plants are commonly consumed as fruits and vegetables worldwide. The aim of these experiments is to determine whether the beneficial trait of the PRS mutation can be applicable to other crops.

In this research, we have applied the RA^{PTR} or the RA^{WT} strain by spraying low abundance to *B. chinensis*, *C. melo* and *C. sativus* cultivated in growth chambers. Under normal agricultural crop management, the addition of microbial inoculants is approximately 10⁸ CFU. This is due to the presence of a variety of biotic and abiotic factors in the complex natural environment, where the ability of microbial inoculants to colonize tends to be significantly reduced. As shown in Fig. 6, the average fresh weight per plant of *B. chinensis* seedling sprayed with low abundance (4×10⁴ CFU/mL) of the RA^{PTR} strain on the 24th day of growth was 36% higher than that of the RA^{WT} strain and 30% higher than that of the control group. The average fresh weights per plant of *C. melo* and *C. sativus* seedlings sprayed with low abundance of the RA^{PTR} strain were 42% and 18% higher than that of the RA^{WT} strain on the 24th day of plants growth, respectively. Furthermore, according to the reviewer 2's suggestion, we have conducted a study to investigate whether the PRS mutation can promote plant yield under agriculturally relevant growth conditions. *Brassica chinensis* (Pak Choi) has been cultivated in a standard Chinese agricultural solar greenhouse. The RA^{PTR} or the RA^{WT} strain was applied by spraying the similar low abundance as the growth chamber. *B. chinensis* was harvested on the 40th day of growth, reaching an optimal size for the market. The colonized cell number of the RA^{PTR} strain was higher than that of the RA^{WT} strain during the *B. chinensis* growth period. And the average fresh weight per hundred plants sprayed with the RA^{PTR} strain (1.82 kg/hundred plants) was 24% higher than that of the RA^{WT} strain (1.47 kg/hundred plants) (Fig. 6 e-g). We highlighted the description of experiments conducted on *B. chinensis* in manuscript on Pages 14-15, Lines 390-398.

We agree with the comment that the genetically engineered microbial strains could not be used for agriculture. We would like to highlight the broader impact of our research. We were able to identify novel metabolic traits highly beneficial for agricultural applications. Prospective developments in the area may include the implementation of targeted enrichment studies and/or experimentally lab evolution (ALE) studies for the isolation or selection of traits with similar characteristics. We also envision that such traits can be implemented in closed cultivation systems, such as aquaponics.

In summary, this manuscript lacks impacts sufficient for publication in *Nat Communications*.

Response: Thank you for all your comments and suggestions! We would like to show that our research provides new insights into plant-microbiome interactions. Here is a summary of our findings that we believe can be of great interest to scientists across different research areas:

Methylobacterium/Methylobacterium stands out as one of the most extensively studied methylotrophs and serves as a significant mutualistic partner that promotes plant growth in the phyllosphere. Although methylotrophic capabilities of *Methylobacterium/Methylobacterium* strains have been recognized to play a role in the colonization process for decades, there are still gaps in our understanding of the mechanisms that enable efficient utilization of methanol in the phyllosphere. Plant-derived methanol is a volatile, one-carbon (C1) compound and its concentration follows the diurnal cycle, varying from trace amounts to tens of millimoles between different stages of plant growth. Identification of cellular factors and pathways that enhance *Methylobacterium/Methylobacterium* growth under fluctuating methanol inputs are vital to developing probiotic supplements that advance establishment of populations with predictable performance for sustainable agriculture.

Current research reports a change in the metabolism of *Methylobacterium/Methylobacterium* strains that improves growth at low methanol concentrations. This change is advantageous for phyllosphere colonization, plant growth, and yield. The findings result from comprehensive research, including a laboratory evolutionary experiment and targeted genetic modifications. The result is a strain that grows better under low methanol concentrations. Genomic resequencing, mutation and cultivation experiments, and metabolites and transcripts analysis were performed to identify and validate the genetic basis and underlying metabolic changes for the improved performance achieved through a single nucleotide mutation in PRS. This PRS mutation establishes a trade-off between nucleotide anabolism and central C1 assimilatory metabolism. It redirects limited C1-carbon resources towards biomass synthesis by up-regulating a non-essential phosphoketolase pathway and facilitates C1-utilization efficiency at low methanol concentrations. These newly acquired traits, characterized by superior plant colonization capabilities, exert a remarkable influence on plant growth and yield, even when applied in low abundance following spray applications under agriculturally relevant growth conditions in agricultural solar greenhouses. The newfound function of PRS is not an isolated phenomenon, it is prevalent across numerous species of bacteria.

The significance of our research is in the demonstration of novel connections between methylotrophic metabolism and cell growth at low methanol concentrations.

The PRS is identified to be an effective metabolic valve, which can promote the establishment of novel *Methylobacterium/Methylorubrum* traits with beneficial impacts on phyllosphere colonization, plant growth and yield. The resulting novel insights may enable the direct selection of *Methylobacterium/Methylorubrum* as a promising plant probiotic. This could have positive effects on sustainable agriculture and one-carbon sequestration.

Another comment:

The first part with RuMP pathway-introduced strains ends with negative results, and this section is very confusing for readers. The authors should remove on the topic or refer to only briefly.

Response: We appreciate your comment. The manuscript was modified according this comment and suggestions by reviewer 2 on this part in the text on Page 5.

Reviewer #2 (Remarks to the Author):

Zhang and colleagues report here about a change in the metabolism of *Methylobacterium* and *Methylorubrum* strains that leads to improved growth at low methanol concentrations, which turns out to be advantageous for phyllosphere colonization. Their findings result from comprehensive research, beginning with a laboratory evolution experiment along with targeted genetic modifications, resulting in a strain with improved growth under low methanol concentrations. This is followed by genomic resequencing and mutation/cultivation experiments to identify and validate the genetic basis and the underlying metabolic changes for the improved performance.

Next, the effect was proven to exist in further strains of the *Methylobacterium/Methylorubrum* group when introducing the same type of mutation. In the last part of the work, the authors show a positive effect on phyllosphere colonization under climate chamber conditions, which is accompanied by increased plant seedling growth. This leads the authors to the conclusion that the identified mechanism can lead to improved plant colonization and represents a lever for more effective phyllosphere colonization of this microbial group, which can go along with improved plant growth.

I consider this an extensive and solid work with very interesting findings. However, I have one major criticism, which is related to the discussed implications the findings may have for crop management. As many studies, this work ends by showing positive effects on plant growth with plants grown in a climate chamber. However, as also stated by the authors, the major challenge when studying plant growth promoting effects of microorganisms is the transfer of lab-based observations to the field. This important validation step is not included in the present work. Even though the results regarding the role of the altered PRS enzyme for efficient plant colonization are convincing, the usefulness for improving the colonization efficiency of *Methylobacterium/Methylorubrum* strains as biologicals under field conditions remains

largely open. Under field conditions, we can expect a stronger competition between *Methylobacterium*/*Methylorubrum* strains with other phyllosphere colonizers. Climate chamber plants usually host a substantially lower abundance of bacteria compared to plants grown under field conditions. Consequently, other resources than carbon may be the limiting factor for methylo-trophs under field conditions and define the population size of the methylo-trophs under these conditions. That the altered traits will indeed lead to an increased population size under such conditions demands further validation. This is further substantiated by the results shown in Fig. 6d, where the differences between wildtype and mutant appear to decrease with increasing plant age. Therewith, the relevance for the improvement of the plant growth potential remains also rather unclear at this stage. The authors should either discuss this much more clearly and be more careful regarding the implications for crop management, or provide further evidence.

Response: Thank you very much for your kind words and support!

We agree with your concerns. In agriculture, crop management are usually conducted in the fields or in solar greenhouses. We have conducted a study to investigate whether the PRS mutation promotes the yield of *Brassica chinensis* (Pak Choi), a common leafy vegetable in China, Asia, Europe, and America, in a standard Chinese agricultural solar greenhouse under agriculturally relevant growth conditions from Dec 24, 2023, to Feb 03, 2024. The results and methods have been added to the text on Pages 14-15, Lines 383-398 and Page 28, Lines 777-791. Briefly, *Brassica chinensis* has been cultivated in a standard Chinese agricultural solar greenhouse. The RA^{PTR} or the RA^{WT} strain was applied by spraying the similar low abundance as the growth chamber. *B. chinensis* was harvested on the 40th day of growth when it had reached the appropriate size for the market. As observed in growth chamber study, the cell number of the RA^{PTR} strain colonizing on *B. chinensis* phyllosphere was significantly higher than that of the RA^{WT} strain during the *B. chinensis* growth period (Fig. 6d) The average fresh weight per hundred plants sprayed with the RA^{PTR} strain (1.82 kg/hundred plants) was 24% higher than that of the RA^{WT} strain (1.47 kg/hundred plants) (Fig. 6). We have also modified the discussion section to be more careful about the implications for crop management in the manuscript on Pages 17-18, Lines 478-485.

Besides this point, other issues should be addressed prior to publication.

The applied statistical tests are not always adequate. Even if the results are convincing and will most likely not change, the statistical analysis should be done correctly. The authors report that they performed t-tests in Excel. Considering that they often analyzed CFU numbers, data are not necessarily normally distributed and may request log-transformation. Was this evaluated? Further, the data in Fig. 5d and 6d request ANOVA with posthoc tests, not multiple t-tests.

Response: We appreciate the reviewer's valuable suggestion. Data in Fig. 5a, b, c, d, f has been all re-analyzed, and log-transformation has been performed before t-tests analysis in Excel. P-values have been corrected in the figures. Data in Fig. 5e, 6d and 6e have been analyzed by ANOVA with posthoc tests in PASW Statistics 18, and *p*-values have been added in the figures. As the reviewer predicted, all the results were

not changed.

It is said that all results are reported based on at least three replicates, often indicated as “n larger or equal to three”. This is fairly unspecific and can in principle be any number larger than three. Please specify the min and max number of replicates of a dataset if not uniform. Besides, the information about replicates given in figure legends appears to be inappropriate in some cases.

Response: The results are based on a minimum of three replicates and a maximum of six replicates. Any non-uniform replicates have been corrected to “ $3 \leq n \leq 6$ ” in the figure legends, and any uniform replicates have also been corrected.

As for Fig. 5e and 6d, the information about replicates seemed inappropriate. We have changed “n = 4 or 3 biologically replicates” to “n=4 or 3 independent biological subgroups”. As the data was obtained as follows: each experimental group was divided into three or four subgroups, five leaves were randomly selected from different plant individuals in each subgroup and were mixed to form a sample, resulting in three or four independent biological subgroups. The colonized cell numbers (gene copy numbers) of the *M. extorquens* AM1 and *Mb. radiotolerans* JCM2831 strains on each subgroup were quantified using qPCR, which allowed for the calculation of the colonized cell numbers per gram of fresh weight of the leaves. We have added this detail in Method and Materials on Page 27, Lines 748-751.

The authors created complex figures with several diagrams, which are very small in size. It is very hard and in part impossible to read the text in these figures on a print out or without substantial page enlargement, especially when diagrams are shown as inserts. Even though articles are nowadays mostly read as pdfs, where figures can be enlarged, a certain size should be provided to enable a reader to quickly grasp and understand the figures. Authors should follow the guidelines.

Response: Thank you for informing us. We have followed the guidelines to modify the figures layout, size dimensions and resolution rates.

The authors should refer correctly to the genera *Methylobacterium* and *Methylorubrum*. Upon reclassification some years ago, these are two different genera and they studied members of both.

Response: Thank you for the comment. The genera name “*Methylobacterium*” and “*Methylorubrum*” has been corrected in the whole manuscript, “*Mb*” refers to “*Methylobacterium*”, and “*Mr*” refers to “*Methylorubrum*”.

Further, the species names of *M. radiotolerans* and *M. oryzae* are frequently misspelled. This needs a systematic correction including their mentioning in text, figures, images and tables (incl. supplementary material).

Response: Thank you very much for the correction. The misspelled “*M. radiotolerans*” and “*M. oryzae*” have been all corrected throughout the manuscript including the supplementary material.

Besides, the names of the strains should be given that were used in this work. This applies to *M. radiotolerans* and *M. oryzae*.

Response: The strains names of *Mb. nodulans* ORS2060, *Mb. radiotolerans* JCM2831, and *Mb. oryzae* CBM20 have been added in the text.

The authors state that some findings and conclusions apply to methylotrophic Alphaproteobacteria, but this claim cannot be made, because they only studied a specific clade, i.e. *Methylobacterium/Methylobacterium*. Besides these, other methylotrophic Alphaproteobacteria exist, which were not part of this study.

Response: Thank the reviewer for pointing out the error. “methylotrophic Alphaproteobacteria” has been replaced by “*Methylobacterium/Methylobacterium*” on Page 10-11, Line 277, 281, 284, and 302.

The manuscript should be proof-read by a native speaker. There are a couple of unclear statements and different grammatical issues. This is somewhat surprising considering that two co-authors are affiliated to Universities in USA and are perhaps native speakers. If not, the manuscript should undergo professional proof-reading. I highlight some issues below to demonstrate the problem, but do not consistently point out all issues.

Response: We appreciate your suggestion. We have invited a native speaker to proofread the entire manuscript.

Some abbreviations are introduced twice in different sections (e.g. ALE). In figure legends, it would be more intuitive to give abbreviations/names in brackets rather than the corresponding explanations (applies to strain derivatives).

Response: Based on your suggestions, we have revised all of these.

Specific comments:

Abstract:

l. 32: rephrase; *Methylobacterium* is not a microbiome, but part of the phyllosphere microbiome

Response: We thank the reviewer for pointing this out. We agree with the comment very much. As the word limit for abstracts (which are required to be 150 or less), we have removed this sentence.

l. 34: it is not proven that *Methylobacterium* is the predominant symbiotic partner of plants in the phyllosphere; it has been reported to be a dominant colonizer of the phyllosphere and strains have been demonstrated to be able to promote plant growth, but whether this genus is indeed the “predominant symbiotic partner that promotes plant growth within the phyllosphere” is not known in general and under field conditions. Rephrase and be more precise regarding this statement.

Response: Thank you very much. We agree with the comment. As the word limit for abstracts (which are required to be 150 or less), we have removed this sentence.

l. 39-40: You state here that you worked with *M. extorquens* AM1 strains being capable of methanol utilization via a heterologous ribulose monophosphate pathway. Can you be sure that this pathway was indeed used? You observed that the plasmid carrying RuMP genes could be removed (after experimental evolution) without problems. This questions whether the pathway was needed at all. Besides, this information is not given in the results section.

Response: We appreciate your valuable suggestion. The introduced RuMP cycle only weakly contributed to methanol assimilation (Supplementary Fig. 1b), as the strain with interrupted *hprA* did not grow on methanol while with the introduced RuMP cycle it grew on methanol slightly. The evolved strains displayed robust growth on standard cultivation condition (i.e., 120 mM methanol) because of the restoration of the native serine cycle, rather than relying on the RuMP cycle for methanol assimilation (Supplementary Fig. 4 and Fig. 5). We have simplified the sentence in the abstract section to “A set of strains originating from *Methylobacterium extorquens* AM1 were subjected to evolutionary pressures to thrive under low methanol conditions.” (Page 2, Lines 35-37). We have also changed the results section to make it clearer on Pages 5-6, Lines 130-135.

l. 41: You write here that the serine cycle was restored, but it is not clear to the reader that it was missing or defect at this point of the abstract.

Response: Thank you for the comment. This sentence in the abstract section has been simplified (Page 2, Lines 35-36). In the results section, additional results related to the serine cycle have been included (Supplementary Fig. 4).

l. 43: add “the” and “gene”, i.e. “in the phosphoribosylpyrophosphate synthetase (PRS) gene was”

Response: The sentence has been corrected (Page 2, Line 37).

l. 50: “offer”, not “offered”

Response: The sentence has been corrected (Page 2, Line 44).

Introduction:

l. 84: the species *M. extorquens* is not necessarily a representative of the *A. thaliana* leaf microbiota, this only applies to specific strains of this species. Add strain name to specify your statement.

Response: The strain name “PA1” has been added (Page 4, Line 81).

l. 96: you probably mean plant growth promoting here, not plant promoting.

Response: Yes, thank you! “plant-promoting” has been corrected to “plant growth-promoting” (Page 4, Line 93-94)

Results:

l. 111: “more efficient” requests a comparative statement; part of the comparison is missing here; it needs to be stated to which alternative pathway the efficiency of the

RuMP pathway is compared.

Response: The sentence has been changed to “a more efficient metabolic route for carbon assimilation compared to the native serine cycle” in the text (Page 5, Lines 117-118).

l. 115: please add a short sentence to say which genes of the RuMP genes were introduced and how.

Response: “Simultaneously, genes of the RuMP cycle from *Bacillus methanolicus* MGA3, namely *phs* and *phi*,^{29,30} were introduced by overexpressing them on plasmid pCM80” has been added in the sentence (Page 5, Lines 121-123).

l. 128-130: two points to this statement: 1.) in case all SE lineages activated the mentioned hypothetical protein, I would point this out more clearly, because the presented results show different performance of different strains; i.e.: add “all” 2.) it would be useful to add a short sentence explaining how this was proven (based on which approach).

Response: Thank you for the suggestion!

1.) The relevant sentence has been corrected (Page 5, Line 132).

2.) To prove that the hypothetical protein META1_3141 performs the function of hydroxypyruvate reductase, it was expressed in *E. coli* BL21 (DE3) and purified. The activity of META1_3141 on hydroxypyruvate was determined by product analysis using GC-MS. It was shown that META1_3141 can catalyze the conversion of hydroxypyruvate to glycerate, which is the action of hydroxypyruvate reductase. The values of *K_m* of META1_3141 were much lower than those reported for hydroxypyruvate reductase. Accordingly, we have added a short sentence in the text (Pages 5-6, Lines 130-135) and added figures in the supplemental material as Fig. 4.

a

b

c

d

K_m (NADH) (mM)	META_3141	1.38
	HPRA	0.04 ^{Reference76}
K_m (NADPH) (mM)	META_3141	0.42
	HPRA	0.06 ^{Reference76}

l. 131: add the, i.e. “in the plant phyllosphere”

Response: The sentence has been corrected (Page 6, Line 137).

l. 135: please specify that this was done via overexpression on a plasmid.

Response: “hprA was reintroduced to the SE strains” was replaced by “hprA was reintroduced to the SE strains via overexpression on plasmid pCM80” (Page 6, Line 142).

l. 137 and elsewhere (especially figure legends): what does “typical methanol” mean? For bacteria under natural conditions 120 mM may not be typical at all; I recommend referring to “standard cultivation conditions” or something alike.

Response: Thank you very much for the comment. In the text, “typical methanol” has been all replaced by “standard cultivation conditions” as the reviewer’s suggestion.

l. 145: here and elsewhere, please state correctly “on low methanol concentrations”; i.e. add the word concentration

Response: All these information has been added in the text.

l. 156 correct by writing “to an asparagine residue”

Response: “to asparagine residue” has been corrected to “to an asparagine residue” (Page 7, Line 165).

l. 177-180: The observation reported in this statement cannot be seen in Fig. 2e. First, I see a clear difference also in the presence of 1.2 mM methanol, but at the same time, data comparison is difficult, because the growth performance under higher methanol concentrations is illustrated based on differences in OD, while the performance under the low concentrations is illustrated based on CFU counts. I don’t understand why different types of data are shown for different concentrations, this should be uniform to allow direct comparison of responses to all conditions. This may explain my disagreement with your statement?

Response: Thank you for the comment. The growth performance under the low concentrations of 0.5 mM and 1.2 mM has been illustrated based on differences in OD (Fig. 2e, please also see below). The growth trends were in consistent with those illustrated based on CFU counts. And the data based on CFU counts has been removed. The description of this result has been revised into “The maximum OD₆₀₀ was increased by 7.0% to 26.7% compared to the AM1^{WT} strain grown on methanol from 0.5 to 15 mM methanol (Fig. 2e)” (Page 7, Lines 185-187).

l. 245: It would facilitate reading if the authors would briefly point out the role of *xfp2* here again.

Response: The sentence of “The *xfp* gene plays a pivotal role in the PKT pathway by catalyzing the conversion of F6P or xylulose-5-phosphate (X5P) into erythrose-4-phosphate or glyceraldehyde-3-phosphate and acetyl-P (Trends Biotechnol. 2022, 40, 149-165)” has been added here in the text (Page 10, Lines 263-265).

l. 251: Reference should be to Fig 3e here, not 3f.

Response: This error has been corrected (Page 10, Line 272).

l. 259: It would be valuable to provide information about the identity of all 64 strains to know which ones were different, e.g. in supplement.

Response: Thank you very much for the comment! Supplementary Fig. 12 (please see below) was added to provide information on the identity of all strains belonging to *Methylobacterium/Methylobacterium*.

1. 289: Do the authors refer to leaf fresh weight or plant fresh weight here?

Response: Herein, “fresh weight” refers to “leaf fresh weight”, and “fresh weight” has been replaced by “leaf fresh weight” in the whole text.

1. 300: Do the authors know for sure that the mutation does not impair the growth of *M. extorquens* in the phyllosphere? Has this been experimentally proven? If so, provide reference.

Response: The following references are now provided:

1) Genetic characterization of the carotenoid biosynthetic pathway in *Methylobacterium extorquens* AM1 and isolation of a colorless mutant. Appl Environ Microbiol. 69, 7563-7566 (2003), as reference 41 in the text.

2) Establishment of CRISPR interference in *Methylobacterium extorquens* and application of rapidly mining a new phytoene desaturase involved in carotenoid biosynthesis. Appl Microbiol Biotechnol. 104, 4515-4532 (2020), as reference 33 in the text.

In these two studies, it has been found that the deletion of the *crtI* gene does not impair the growth of *Mr. extorquens* under standard cultivation conditions; The reference was added on Page 11, Line 306. In our research, it has been demonstrated that the colonization ability of the YAIP strain (the *crtI* deleted strain) in the phyllosphere was as competitive as the WT strain (Fig. 5d). This indicates that the deletion of *crtI* does not impair the growth of *Mr. extorquens* in the phyllosphere. To avoid misleading readers, the sentence has been corrected into “This mutation does not impede *Mr. extorquens* growth under standard cultivation conditions” (Page 12, Lines 323-324).

l. 307-308: You may want to specify the purpose of this experiment here.

Response: The purpose of this experiment is “to assess the colonization efficiency of the AM1^{PTR} strain in non-sterile conditions.”. This information is now presented in the main text, on Page 12, Lines 333-334.

l. 308-310: The sentence is confusing and probably not correct, because in Fig. 5e results from foliar application are shown, while it is said here that seeds were treated.

Response: Thank you very much, and the error has been corrected into “*A. thaliana* seeds were planted in non-sterile soil and watered with tap water. After growing for 7 days, *A. thaliana* seedlings were foliar sprayed with either the AM1^{PTR} strain or AM1^{WT} strain. The control group was foliar sprayed with 10 mM MgCl₂.” in the text as your suggestion on Page 12, Lines 334-337.

l. 320-324: phrasing unclear.

Response: We have revised the sentences as follows: Plant leaves can release various types of multi-carbon sources, such as oxalate and succinate. The central metabolic pathway for utilizing multi-carbon sources by *Methylobacterium/Methylobacterium* strains differs from methylo-trophy^{42, 43}. The colonization of the AM1^{PTR}04 strain suggested that the activation of the PKT pathway might also influence the efficiency of *Methylobacterium/Methylobacterium* strains in utilizing these multi-carbon sources in the phyllosphere. (Page 13, Lines 348-353).

l. 333: “visible abundance” does not appear to be a meaningful phrase; what would that be?

Response: “low visible abundance” has been replaced to “low-abundance” (Page 12, Line 362).

l. 335: “open climatic chamber”? What do the authors mean with this?

Response: The phrase “open climatic chamber” refers to a non-sterile growth chamber. To make it clearer, “in the open climatic chamber” is replaced with “in a non-sterile growth chamber” in the text (Page 14, Line 367).

l. 338: specify what the control group was.

Response: The control group was treated by 10 mM MgCl₂. In the text, “the control group treated with 10 mM MgCl₂” has been added (Page 14, Line 370).

Discussion:

l. 416: add reference to support this statement

Response: Reference of “Deep discovery informs difficult deployment in plant microbiome science, Cell. 2023 Oct; 186: 4496-4513.” has been added as reference 55 here (Page 17, Line 467).

l. 429-431: not clear how this can be realized, because to my understanding it needs genetic engineering of the Methylobacterium strains to have the observed improved performance.

Response: Thank you very much for the correction. In order to avoid misleading information, we have deleted this sentence.

Methods:

Thank you for all the suggestions. The addition or change are as follows:

The genomic sequencing part is not explained with sufficient detail to reproduce.

Response: Detailed description of genome sequencing method has been added in the text (Pages 21-22, Lines 594-512).

Check the correct formatting of gene names (e.g. l. 557, 561 and possibly elsewhere).

Response: Formatting of gene names have been corrected in the whole text.

qPCR cycling conditions should be described to enable others to reproduce it.

Response: The detailed qPCR cycling conditions have been added to the text as follows: “The qPCR cycling conditions were: 1. 95 °C for 5 min; 2. 95 °C for 10 sec; 3. 60 °C for 34 sec; 4. go to 2 for 40 cycles; 5. 95 °C for 15 sec; 6. Melt curve, 60 °C to 95 °C (increment 0.15°C/s) for 1 sec.” (Page 22, Lines 622-625).

l. 595-598: sentence grammatically incorrect; rephrase

Response: This sentence has been rephrased as follows: “The *Mr. extorquens* strains were cultured on methanol at a concentration of 10 mM. 20 mL of cultures at the exponential phase (OD₆₀₀ of 0.4 ± 0.05) were rapidly harvested by vacuum filtration through 0.22-µm nylon membrane filters, and the collected strains were promptly quenched with liquid nitrogen, following established procedures” (Page 24, Lines 664-667).

l. 652: provide protocol for *A. thaliana* seed sterilization or give an appropriate reference

Response: We have added a reference of “A proteomic study of *Methylobacterium extorquens* reveals a response regulator essential for epiphytic growth” (Proc Natl Acad Sci U S A. 2006 Aug 29;103(35):13186-91.) as reference 72 (Page 26, Line 736).

l. 661 and 673: how to know that this number of CFUs was applied per plant? Is it the theoretical maximum number? Not all cells will reach the leaf surface when spraying I assume. Better to provide in addition information on the cell density of the suspension that was applied and the amount of liquid that way sprayed per plant.

Response: Thank you very much for the correction. The number of CFUs applied per plant is the theoretical maximum number as not all cells reach the leaf surface during spraying. The texts have been corrected into “*A. thaliana* plants were allowed to grow for 7 days prior to foliar spraying with 1 mL of culture liquid per plant. The culture liquid contained either the PRS mutant strain or its WT counterpart. Specifically, the culture liquid consisted of 1 mL of a 10 mM MgCl₂ solution containing 10⁶ CFUs of *Mr. extorquens* cells.” as your suggestion (Page 27, Lines 743-747, 768-770).

Figures/tables:

l. 897: correct formatting of WT

Response: The formatting of WT has been corrected to “AM1^{WT}01.1” (Page 40, Line 1052).

Fig. 3e: Is “NA” correct (not analyzed, not applicable, not available?); shouldn't it be “n.s.” (not significant).

Response: Here “NA” was abbreviated for “not analyzed”. To make it clearer, the sentence, “NA” is abbreviated for “not analyzed”, has been added in the legend (Page 42, Line 1095).

Fig. 4b: It could have been interesting to evaluate whether the 34 strains represent specific subgroups within the *Methylobacterium*/*Methylorubrum* diversity, but as only the 34 strains are shown, I don't see the relevance of this tree. But this appears not to be the case, so an extended tree will probably not add anything, so I recommend removing it. Further, the legend does not indicate based on which gene(s) the tree was calculated and based on which algorithm, not does it explain the red dots behind some strains. More important would be the identity of all strains included in the genomic analysis in the supplement.

Response: Thank you for the comment. Information about the identity of all strains belonging to *Methylobacterium*/*Methylorubrum* has been provided. Fig. 4b has been deleted and the identity of all strains included in the genomic analysis has been added as Supplementary Fig. 12. The genes based on which the tree was calculated and the algorithm used have been added to the legend of Supplementary Fig. 12.

Fig. 5 a-c: It is unclear what replicate 1 and replicate 2 means in this context, assuming

that each dot is representing the CFU determined on one plant individual, which would in my understanding represent a replicate. Besides, the statistics with such low p-values were probably not done based on two replicates only. Either the definition of replicates is inappropriate, the statistics, or both.

Response: Thank you for the comment. To strengthen the results, we have conducted two independent experiments. Replicate 1 and Replicate 2 refer to independent experiment 1 and independent experiment 2, respectively. The replicates from two independent experiments have been labelled as independent experiment 1 and independent experiment 2 in the figure. Plant individuals in independent experiment 1 were shown as n1, and plant individuals in independent experiment 2 were shown as n2 in the figures.

Fig. 5e: this figure reports qPCR results, the y-axis labelling with CFU is incorrect. It should report gene copy numbers, if a specific gene was targeted. Otherwise, adjust accordingly.

Response: Thank you very much! We agree with the comment. The results reported gene copy numbers which reflect colonized cell numbers, rather than viable cell numbers (i.e. CFU). We have changed the y-axis labelling with gene copy numbers and modified the text accordingly. And the method description was added in detail to the text (Page 27, Lines 759-763).

Fig. 6 b and c: at which day/plant age were these images taken and the data determined?

Response: In this figure, the images and the data were taken 12-day post foliar spray when the plant was 24-day old. "Pictures and data were taken at 12-day post foliar spray (plant age of 24-day)" has been added in the legend (Page 46, Line 1138-1139).

Fig. 6: legend unclear by stating that data are based on n = 3 biological replicates, I see already 6 plant individuals in the photographs and many dots in panel c.

Response: The six plant individuals in Fig. 6b was randomly selected from each of three experimental groups (Control/RA^{WT}/RA^{PTR}). The number of biological replicates in Fig. 6c was indicated in the figure, $40 \leq n \leq 50$, and each dot represented a plant individual. As for Fig. 6d, the data was obtained as follows: each experimental group were divided into three subgroups, five leaves were randomly selected from different plant individuals in each subgroup and were mixed to form a sample, resulting in three independent biological subgroups. The gene copy numbers (colonized cell numbers) of the *Mb. radiotolerans* JCM2831 strains on each subgroup were quantified using qPCR. We have added this detail in Method and Materials on Page 27, Lines 748-751. We have also changed "n = 3 biological replicates" to "n=3 independent biological subgroups" in the legend.

Files with supplementary figures and tables should be declared as such, in the file name (if possible) and within the file. For the tables given in excel files, it would be useful to add the table header within the respective data file rather than in a separate document.

Response: We have modified this part as your suggestion.

Fig S2: I don't see a need to show Fig. 2a, because this did not result in any strains that were further studied here.

Response: Thank you for the comment. Fig S2a has been removed.

Fig. S3 would profit from a bit more detailed legend.

Response: We have added more information to make it clearer.

Table S6: please provide information about target genes and/or target organism if specific.

Response: We have added the information about target genes and target organisms.

Reviewer #3 (Remarks to the Author):

Zhang and collaborators demonstrate that the partial loss of the phosphoribosylpyrophosphate synthase (PRS) function discretely but significantly improved *Methylobacterium* growth in conditions of limited carbon sources, and this loss enhanced plant colonization. This finding is original and of great interest to both fundamental and applied scientific communities.

We greatly appreciate your positive feedback and great suggestions!

Comments:

1. Although methanol was the substrate used for the discovery that PRS is a metabolic valve that enhances *Methylobacterium* phyllosphere colonization, this process might not be restricted to methanol as shown in Figure 1. The effect of the PRS mutation on *Methylobacterium* growth, a facultative methylotroph, was observed for methanol and other multi-carbon sources, each studied alone as the only carbon source, but no combination of limiting amounts of different carbon sources was assessed together in this study (although low concentration of methanol and multi-carbon compounds are found in the phylloplane).

Response: We appreciate your valuable suggestions! The growth of the AM1^{WT} and AM1^{PTR} strains on the combined carbon sources (0.5 mM Methanol, 0.1 mM ethanol, 0.1 mM acetate, 0.1 mM oxalate, 0.1 mM pyruvate, and 0.1 mM succinate) has been compared as suggested. Similar to the low concentration of the sole carbon source, the AM1^{PTR} strain showed growth advantage over the AM1^{WT} strain on combined carbon sources at low concentration. The result has been added in Fig. 2 as Fig. 2j. The description of this result has been added in the text on Page 8, Lines 109-211.

2. A high methanol concentration of 120mM is not a ‘typical’ concentration used in labs working with *Methylobacterium* strains. The point here is that this concentration provides methanol in excess and is not growth-limiting. Revise the text in consequence throughout the manuscript (L117, L124, etc.).

Response: Thank you for making this good point. This suggestion has also been made by reviewer 2. In response to the suggestions, we have replaced “typical concentration” with “standard cultivation conditions” throughout the manuscript.

3. Few *Methylobacterium* species have been described as strong symbionts, which may not be the case of the species studied in this work (except for *M. nodulans*). To avoid misleading the reader, consider modifying the text in the abstract (L34).

Response: We thank the reviewer for pointing this out. We agree with the comment. As the word limit for abstracts (which are required to be 150 or less), we have removed this sentence. In addition, we have changed “symbiotic relationships” to “mutualistic relationships” on Page 2, Line 35.

4. Do the strains used in this study have the Δcel mutation that prevents cell aggregation in lab cultures? If not, this may affect DO and CFU proportionality measurements with biomass. Did the cells aggregate in this study?

Response: The strains used in this study were not Δcel mutated. In this work, *Methylobacterium/Methylorubrum* strains were usually grown at low methanol concentrations. As a result, the maximum OD₆₀₀ was less than 0.7, and we did not observe the cell aggregation. In addition, six replicates were used to determine OD and CFU in order to obtain accurate biomass.

5. For the study of the PRS mutation effect on *A. thaliana* leaf colonisation from seed inoculation, the experiments were carried out with *M. extorquens* AM1. Why was *M. extorquens* AM1 chosen, rather than PA1, which is known to be a better colonizer?

Response: Thank you for the comment. The aim of this study was to investigate whether the average colonization ability of *Methylobacterium/Methylorubrum* could be improved after the introduction of the PRS mutation. As shown in Fig. 5a and b, *Mr. extorquens* AM1 colonized less than PA1 bacteria, which is consistent with PA1 bacteria being able to colonize *Arabidopsis* better (also as your point), indicating colonization of *Mr. extorquens* AM1 needs to be improved. Therefore, *Mr. extorquens* AM1 was selected for further investigation of the effect of colonization in the presence of a competing strain (Fig. 5d) and in a non-sterile growth chamber (Fig. 5e).

In Fig. 5e, provide the qPCR data for the detection of *M. extorquens* AM1 population on *Arabidopsis* leaves, in addition to viable count cells. How can CFU be safely assigned to *M. extorquens* AM1? How were the AM1 specific primers validated?

Response: Thank you very much for the comment. This is also pointed out by Reviewer 2. The gene copy numbers obtained by qPCR reflect colonized cell numbers rather than viable cell numbers (i.e. CFU). We have corrected this error. We have changed the y-axis labelling with gene copy numbers and modified the text accordingly. And the

method description was added in detail to the text (Page 27, Lines 759-763).

We selected a unique genomic region in *Mr. extorquens* AM1 (GenBank, CP001510.1, 1631527..1632060) which was not present in other bacterial strains, including *Mr. extorquens* PA1. Specific primers were designed to amplify this unique genome region.

6. After spraying *M. radiotolerans* WT strain or PRS mutant on different plant seedlings, how do you assign CFU to *M. radiotolerans* and distinguish them from native epiphytes (Fig 6d; L342-344)?

Response: The method used to assign colonized cell numbers to *Mb. radiotolerans* JCM2831 was the same as that used in Fig. 5e. Specific primers for qPCR were designed using a unique genomic region in *Mb. radiotolerans* JCM2831 (GeneBank: CP001001.1, 1221888..1223114) to distinguish it from other native epiphytic bacterial strains. Notably, it was unable to distinguish sprayed *Mb. radiotolerans* JCM2831 from native epiphytic *Mb. radiotolerans*. A control group was set up, which was sprayed with MgCl₂ solution to determine the CFU of native *Mb. radiotolerans*.

Other comments:

Fig. 3c. For clarity, specify in the legend that version 04 indicates a mutated *xfp2* gene.

Response: Thank you for your suggestion. The legend has been changed to "Knocking out the gene *xfp2* (AM1^{PTR} 04) decreased the growth advantage of the AM1^{PTR} strain on low methanol concentrations of 5 mM, 10 mM, and 15 mM, while it did not affect the growth of the AM1^{WT} strain".

Fig. 3e. Why classify "His" under nucleotides?

Response: We have changed "nucleotides" to "nucleotides and amino acids" in Fig. 3e.

Fig. 4a. Specify the "other *Methylobacterium* species"

Response: Here "other *Methylobacterium* species" referred to "all other 61 *Methylobacterium/Methylorubrum* species". The legend has been changed to "SeqLogo schematic showing the conservation of amino acid residues of PRS in all *Methylobacterium/Methylorubrum* species."

Fig. 4b. Specify the identity of all of the *Methylobacterium* strains used in the phylogenetic analysis.

Response: Thank you for the comment. Information about the identity of all strains belonging to *Methylobacterium/Methylorubrum* has been provided. Fig. 4b was moved to Supplementary Fig. 12 as Reviewer 2 suggestion.

Fig. 6b. No size ruler was shown in the different pictures.

Response: We have added size ruler as your suggestion.

Supplemental Fig. 1. Indicate color code

Response: Thank you for the comment. Heterologously introduced genes were

indicated in red, and knocked-out genes were indicated in grey. The color code description is now added.

Supplemental Fig. 3. Correct “*M. radiotolarance*”. What is the color code? Correct the text covered by arrows.

Response: We apologize for the typo. The misspelled “*M. radiotolerance*” has been corrected. The figure was corrected as suggested, and unified the colors in the figure.

REVIEWER COMMENTS

Reviewer #1 (Remarks to the Author):

The authors respond to all of the comments from the reviewers.

I suggest the manuscript can be accepted for publication in Nature Com.

Reviewer #2 (Remarks to the Author):

The authors submitted here a substantially revised version of their manuscript. Concerns were addressed by providing additional data and adding further information to the manuscript. Despite these improvements, some issues could be further improved in my opinion.

On p. 14, l. 378-382 the authors describe the underlying mechanism of plant growth promotion. The mechanism put forward by the authors to explain the stronger plant growth promotion by the modified *Methylobacterium* strains remains with some uncertainty and this should be stated more clearly in my opinion. I can follow the argumentation that higher population sizes of *Methylobacterium*/-*rubrum* have the potential to cause a stronger growth response of plants. The higher population size is well proven. However, whether this is mediated in planta by higher IAA and ACC deaminase levels remains an assumption. The authors showed that IAA and α -ketobutyric acid (product of ACC deaminase) production was not impaired in the mutant strain based on laboratory cultures only. This is an important prerequisite for the proposed mechanism behind the stronger growth promotion, but whether this is indeed the mode of action that explains here the growth promotion along with the larger population size requests further in planta proof. I do not necessarily ask for this proof, but the conclusions about the underlying mechanism could be phrased a bit more carefully. Or at least literature should be more carefully selected and cited that has convincingly proven the mode of action for this particular strain and for both mechanisms in planta.

I recommend that the rationale for forming “independent biological subgroups” for some analyses is briefly explained/justified in the methods section.

I perceive the font sizes in some figures still as too small; they are not readable when figures are printed and digital enlargement of some plots results in blurry lines and hard-to-read text. The authors state that figures were prepared following the guidelines – if so, I will not insist on further improvement – just make them aware that there is room for further improvement.

L. 324: in contrast to the statement given in the text, the results presented in fig. 5 indicate a slight growth impairment of the non-pigmented strain YAIP when colonizing plants compared to a wild type. This is well conceivable, as carotenoids may be important under light-conditions and thus for plant colonization of *Methylorubrum*/-*bacterium*. This is why I asked for clarification on this point. Instead of ignoring the slight difference and referring to references that claim that there are no impairments known (when assessed under standard cultivation conditions), it would be better to explain the observed slight difference in Fig. 5d. At the end, this does not affect the relevant

conclusion that AM1-PTR was more competitive against YAIP than AM1-WT, but provides more transparency.

l. 592: provide a reference for what you call the “SDS method”

I requested to correct the y-axis labeling of figures showing qPCR results and in addition asked for more details regarding qPCR. I was assuming that the authors used a small DNA fragment with the respective target gene/region and flanking regions in their qPCR standard, as it is usually done to estimate absolute abundances. Consequently, I recommended the term “gene copy numbers” for y-axis labeling. However, the authors prepared their qPCR standards from genomic DNA extracted from pure cultures with known cell numbers instead of using a known number of DNA molecules with the target gene; an approach I see here for the first time. This difference can be important when comparing data to those of other studies, because this procedure will be affected by different bias compared to the other approach. Based on this, the term “cell numbers” may indeed be used as y-axis label in this study, because “gene copy numbers” is rather misleading.

Genus names are still not consistently spelled correctly everywhere, e.g. l. 291.

There are likewise still language errors in the manuscript and some unclear statements; e.g. l. 71-75: complicated sentence, l. 283-285: incorrect phrasing. It needs another careful proof-reading to eliminate further mistakes. There are quite often 1-2 per page left.

l. 773: “previously” requests for a reference. The authors probably mean “as described before”?

Reviewer #3 (Remarks to the Author):

The revisions have improved the manuscript. New data have been provided, closer to agriculturally-relevant growth conditions, using a standard Chinese agricultural solar greenhouse, to test for the effect of spraying the leafy vegetable *Brassica chinensis* with either RAprt or RAWt strains. A discrete but significant plant promoting growth effect was observed with the RAprt strain. Thus, the overall impact of this MS has increased.

The reduction of biogenic C1 emissions and sequestration is an important issue to consider and discuss for future sustainable agriculture. In this study, PRS-mutated methylotrophs are proposed to have reduced CO₂ emissions compared to WT strains (see Fig. 1), but what about the C1 emission impact of these plant-promoting methylotrophics on treated plant?

Ribulose-5-phosphate is the substrate of PRS but also of another enzyme whose phosphoribulokinase activity produces ribulose-1,5-bisphosphate shown previously to be central for the regulation of one-carbon assimilation in *M. extorquens* PA1 (ref n°26). This feature could be added to the MS (L161-164) and discussed in view of the new data provided in Fig 2J (L209-211).

In the MS, at the first occurrence of “standard conditions”, can the authors state that in these growth conditions, methanol concentrations (60-120mM) are clearly above growth-limiting concentrations (around 15mM or less).

L70. Replace “symbiotic” with “associated”

Add “bacterial” in front of “traits” in the abstract (L40_42) and in front of “efficient utilization” in the introduction (L67).

AUTHORS RESPONSES TO THE REVIEWERS COMMENTS.

Reviewer #1 (Remarks to the Author):

The authors respond to all of the comments from the reviewers.
I suggest the manuscript can be accepted for publication in Nature Com.

We would like to express our great appreciation for your time and valuable feedback!

Reviewer #2 (Remarks to the Author):

The authors submitted here a substantially revised version of their manuscript. Concerns were addressed by providing additional data and adding further information to the manuscript. Despite these improvements, some issues could be further improved in my opinion.

We greatly appreciate your nice words and valuable suggestions!

On p. 14, l. 378-382 the authors describe the underlying mechanism of plant growth promotion. The mechanism put forward by the authors to explain the stronger plant growth promotion by the modified *Methylobacterium* strains remains with some uncertainty and this should be stated more clearly in my opinion. I can follow the argumentation that higher population sizes of *Methylobacterium*/-*rubrum* have the potential to cause a stronger growth response of plants. The higher population size is well proven. However, whether this is mediated in planta by higher IAA and ACC deaminase levels remains an assumption. The authors showed that IAA and α -ketobutyric acid (product of ACC deaminase) production was not impaired in the mutant strain based on laboratory cultures only. This is an important prerequisite for the proposed mechanism behind the stronger growth promotion, but whether this is indeed the mode of action that explains here the growth promotion along with the larger population size requests further in planta proof. I do not necessarily ask for this proof, but the conclusions about the underlying mechanism could be phrased a bit more carefully. Or at least literature should be more carefully selected and cited that has convincingly proven the mode of action for this particular strain and for both mechanisms in planta.

Response: Thank you for the suggestions! We agree with your comments very much.

The *prs* mutation did not affect IAA and ACC deaminase production in the laboratory cultures in this study. It is only a prerequisite for the proposed mechanism behind the enhanced growth promotion of the *prs* mutant strain in the plant. We were in complete agreement that further in planta evidence was needed to determine whether this was indeed the IAA and ACC deaminase production that explained the plant growth promotion, along with the larger population size of the *prs* mutant strain. Consequently, we have modified the sentences that attempt to propose the mechanism behind the stronger growth promotion to a hypothesis that could be verified in future studies. The sentences have been modified to “The PRS mutation in the RA strain did not seem to affect the synthesis of indoleacetic acid and ACC deaminases under laboratory cultivation conditions (Supplementary Fig. 13). Therefore, we hypothesized that the higher plant growth-promoting ability of the RA^{PTR} strain might be attributed to greater number of strains colonizing the phyllosphere.” (lines 390-394).

I recommend that the rationale for forming “independent biological subgroups” for some analyses is briefly explained/justified in the methods section.

Response: Thanks for the valuable advice. To avoid damaging plant growth, we did not take leaf samples from each plant individual to detect the colonized cell numbers. Instead, in order to reflect the colonized cell numbers as accurately as possible, we divided the plant individuals in each experimental group into three or four subgroups evenly. Five leaves were randomly selected from different plant individuals in each subgroup and mixed to determine the number of colonized cells. The sentence “To avoid damaging plants growth and to measure the colonized cell numbers as accurately as possible” has been added in the revised manuscript (lines 760-761).

I perceive the font sizes in some figures still as too small; they are not readable when figures are printed and digital enlargement of some plots results in blurry lines and hard-to-read text. The authors state that figures were prepared following the guidelines – if so, I will not insist on further improvement – just make them aware that there is room for further improvement.

Response: Thank you for drawing our attention to some shortcomings of the figures. We have enlarged the font size in the figures as the suggestion. We also noticed that some lines and text in the figures became blurred when the file was converted to PDF. It is believed that they would be much clearer with the original image version we submit via the Nature's systems platform.

L. 324: in contrast to the statement given in the text, the results presented in fig. 5 indicate a slight growth impairment of the non-pigmented strain YAIP when colonizing plants compared to a wild type. This is well conceivable, as carotenoids may be important under light-conditions and thus for plant colonization of *Methylobacterium*/-bacterium. This is why I asked for clarification on this point. Instead of ignoring the slight difference and referring to references that claim that there are no impairments known (when assessed under standard cultivation conditions), it would be better to explain the observed slight difference in Fig. 5d. At the end, this does not affect the relevant conclusion that AM1-PTR was more competitive against YAIP than AM1-WT, but provides more transparency.

Response: Thanks for the great suggestion. The sentence “This mutation does not impede *Mr. extorquens* growth under standard cultivation conditions” was changed to “This mutation does not impede the growth of *Mr. extorquens* under standard cultivation conditions^{33,41}; however, it caused slight growth impairment when colonizing plants (Fig. 5d), as carotenoid is considered crucial under light-conditions and thus for plant colonization⁴².” (lines 333-336).

l. 592: provide a reference for what you call the “SDS method”

Response: The reference has been added. (line 604)

I requested to correct the y-axis labeling of figures showing qPCR results and in addition asked for more details regarding qPCR. I was assuming that the authors used a small DNA fragment with the respective target gene/region and flanking regions in their qPCR standard, as it is usually done to estimate absolute abundances. Consequently, I recommended the term “gene copy numbers” for y-axis labeling. However, the authors prepared their qPCR standards from genomic DNA extracted from pure cultures with known cell numbers instead of using a known number of DNA molecules with the target gene; an approach I see here for the first time. This difference can be important when comparing data to those of other studies, because this procedure will be affected by different bias compared to the other approach. Based on this, the term “cell numbers” may indeed be used as y-axis label in this study, because “gene copy numbers” is rather misleading.

Response: The figures were modified as suggested. The y-axis labels of Fig. 5e, 6d, and 6e have been changed to “Cell numbers g⁻¹ FW”.

Genus names are still not consistently spelled correctly everywhere, e.g. l. 291.

Response: We sincerely apologize for the errors. We have carefully corrected the misspelled genus names through the revised manuscript.

There are likewise still language errors in the manuscript and some unclear statements; e.g. l. 71-75: complicated sentence, l. 283-285: incorrect phrasing. It needs another careful proof-reading to eliminate further mistakes. There are quite often 1-2 per page left.

Response: We would like to express our great gratitude for the comments. In order to ensure the high standards of accuracy, we asked a native speaker to undertake a careful proofreading of the manuscript.

l. 773: “previously” requests for a reference. The authors probably mean “as described before”?

Response: “previously” has been changed to “as described before” in the revised manuscript (line 786).

Reviewer #3 (Remarks to the Author):

The revisions have improved the manuscript. New data have been provided, closer to agriculturally-relevant growth conditions, using a standard Chinese agricultural solar greenhouse, to test for the effect of spraying the leafy vegetable *Brassica chinensis* with either RAprt or RAWt strains. A discrete but significant plant promoting growth effect was observed with the RAprt strain. Thus, the overall impact of this MS has increased. We greatly appreciate your nice words and valuable suggestions.

The reduction of biogenic C1 emissions and sequestration is an important issue to consider and discuss for future sustainable agriculture. In this study, PRS-mutated methylotrophs are proposed to have reduced CO₂ emissions compared to WT strains (see Fig. 1), but what about the C1 emission impact of these plant-promoting methylotrophics on treated plant?

Response: Thanks for the suggestion. The PRS-mutated *Methylobacterium/Methylorubrum* may potentially reduce CO₂ emissions during the metabolism, as the CO₂ fixation was enhanced and the activity of formate dehydrogenases was reduced. The mutated *Methylobacterium/Methylorubrum* might also reduce the methanol emission of plants, as the colonized cell numbers were increased, which consequently would consume more methanol emitted by plants. Furthermore, the mutated *Methylobacterium/Methylorubrum* species have been observed to promote plant growth in this study, which could then enhance the CO₂ fixation of plants and likely result in an increased sequestration of CO₂ by treated plants. In this study, we have not yet measured the C1 emission from *Methylobacterium/Methylorubrum*-treated plants, and we will undertake this important evaluation in the near future to contribute to the field of sustainable agriculture.

Ribulose-5-phosphate is the substrate of PRS but also of another enzyme whose phosphoribulokinase activity produces ribulose-1,5-bisphosphate shown previously to be central for the regulation of one-carbon assimilation in *M. extorquens* PA1 (ref n°26). This feature could be added to the MS (L161-164) and discussed in view of the new data provided in Fig 2J (L209-211).

Response: Thanks for the excellent advice. Indeed, we found that ribose-5-phosphate (Ri5P), the substrate of PRS, was misspelled as ribulose-5-phosphate. We are sorry for this error and we have carefully double-checked the spelling of the other metabolites to ensure that no other errors have been made. The sentence “It is essential to note that PRS catalyzes the first rate-limiting reaction, converting ribose-5-phosphate (Ri5P) to 5-phosphoribosyl-1-pyrophosphate (PRPP), a critical step in nucleotide biosynthesis pathways. Additionally, ribulose-5-phosphate (Ru5P), a metabolite interconvertible with Ri5P, is catalyzed by another enzyme, phosphoribulokinase, to produce ribulose-1,5-bisphosphate (RuBP). Previous studies have demonstrated the pivotal role of RuBP in the regulation of one-carbon assimilation in *Mr. extorquens* PA1²⁶.” has been added in the revised manuscript (lines 166-172).

A further investigation is currently underway to determine how the PRS mutation enhances the utilization of limited quantities of multi-carbon sources and whether the PRS mutation is involved in ribulose-1,5-bisphosphate function. We are grateful to the reviewer for the valuable suggestion and will endeavor to incorporate the perspective in our forthcoming work. Accordingly, the sentence “The mechanisms that underlie the enhanced utilization of limited multi-carbon sources, as well as combinations of all carbon sources from C1 to C4, by the PRS mutation, are currently under investigation and will be reported in a subsequent publication.” has been added to lines 282-285.

In the MS, at the first occurrence of “standard conditions”, can the authors state that in these growth conditions, methanol concentrations (60-120mM) are clearly above growth-limiting concentrations (around 15mM or less).

Response: The statement has been added at the first occurrence of “standard conditions” (lines 128-130). Specifically, the sentence “In this study, standard cultivation conditions referred to methanol at a concentration of approximately 120 mM, which were clearly above growth-limiting concentrations (15 mM or less).” has been added after the sentence “The resulting strain, named RS03, initially exhibited minimal growth under standard cultivation conditions (i.e., 120 mM methanol, Supplementary Fig. 1b).”

L70. Replace “symbiotic” with “associated”

Response: “symbiotic” has been replaced with “associated” in the revised manuscript (line 73).

Add “bacterial” in front of “traits” in the abstract (L40_42) and in front of “efficient utilization” in the introduction (L67).

Response: “bacterial” has been added in front of “traits” in line 44, and also in front of “efficient utilization” in line 70 as suggested.

REVIEWERS' COMMENTS

Reviewer #2 (Remarks to the Author):

I don't have further comments on this manuscript.

Reviewer #3 (Remarks to the Author):

The authors revised the manuscript as per the reviewer's comments: the revised manuscript can be accepted.